


# IRIS analyser assessment reveals sub-hourly variability of isotope ratios in carbon dioxide at Baring Head, New Zealand's atmospheric observatory in the Southern Ocean

Peter Sperlich[1], Gordon W. Brailsford[1], Rowena C. Moss[1], John McGregor[1], Ross J. Martin[1], Sylvia Nichol[1], Sara Mikaloff-Fletcher[1], Beata Bukosa[1], Magda Mandic[2], Ian Schipper[3], Paul Krummel[4]

[1]National Institute of Water and Atmospheric Research (NIWA), Wellington, New Zealand
[2]Thermo Fisher Science, Bremen, Germany
[3]Victoria University of Wellington, New Zealand
[4]Commonwealth Scientific and Industrial Research Organisation (CSIRO), Aspendale, Australia

*Correspondence to*: Peter Sperlich (peter.sperlich@niwa.co.nz)

**Abstract.** We assess the performance of an Isotope Ratio Infrared Spectrometer (IRIS) to measure carbon ($\delta^{13}C$) and oxygen ($\delta^{18}O$) isotope ratios in atmospheric carbon dioxide ($CO_2$) and report observations from a 26 day field deployment trial at Baring Head, New Zealand, NIWA's atmospheric observatory for Southern Ocean baseline air. Our study describes an operational method to improve the performance in comparison to previous publications on this analytical technique. By using a calibration technique that reflected the principle of identical treatment of sample and reference gases, we achieved a reproducibility of 0.07 ‰ for $\delta^{13}C$-$CO_2$ and 0.06 ‰ for $\delta^{18}O$-$CO_2$ over multiple days. This performance is within the "extended compatibility goal" of 0.1 ‰ for both $\delta^{13}C$-$CO_2$ and $\delta^{18}O$-$CO_2$, which was recommended by the World Meteorological Organisation (WMO) for studies of regional or urban $CO_2$ fluxes. One goal of this study was to assess the capabilities and limitations of the IRIS analyser to resolve $\delta^{13}C$-$CO_2$ and $\delta^{18}O$-$CO_2$ variations under field conditions. Therefore, we selected multiple events within the 26 day record for Keeling Plot Analysis. This resolved the isotopic composition of end members with an uncertainty of ≤1 ‰ when the magnitude of $CO_2$ signals is larger than 10 ppm. The uncertainty of the Keeling Plot Analysis strongly increased for smaller $CO_2$ events (2-7 ppm), where the instrument performance is the limiting factor and may only allow for the distinction between very different end members, such as the role of terrestrial versus oceanic carbon cycle processes.

Further improvement in measurement performance is desirable to meet the WMO "network compatibility goal" of 0.01 ‰ for $\delta^{13}C$-$CO_2$ and 0.05 ‰ for $\delta^{18}O$-$CO_2$, which is needed to resolve the small variability that is typical for background air observatories such as Baring Head.



## 1. Introduction

Carbon dioxide ($CO_2$) is the single most important anthropogenic greenhouse gas, and it is therefore of critical importance to understand biogeochemical processes controlling atmospheric $CO_2$ levels (IPCC 2021). The isotopic composition of atmospheric $CO_2$ at any time and location is controlled by different carbon cycle processes and can therefore be used to constrain carbon fluxes on a range of spatio-temporal scales. For example, Ciais et al. (1995a; 1995b) have used stable carbon isotope ratio in atmospheric $CO_2$ ($\delta^{13}C$-$CO_2$) in weekly samples from 43 sites to distinguish terrestrial and ocean sink fluxes. Keeling et al. (2017) also used $\delta^{13}C$-$CO_2$ from flask samples to infer changes in water use efficiency of plants with increasing atmospheric $CO_2$ mole fractions. Similarly, the oxygen isotope ratio ratios in $CO_2$ ($\delta^{18}O$-$CO_2$) have been used as a tracer for gross primary production (GPP) of the terrestrial biosphere (Francey and Tans, 1987; Ciais et al., 1997). Using global $\delta^{18}O$-$CO_2$ records, Welp et al. (2011) demonstrated the impact of El Niño Southern Oscillation on the global carbon cycle and provided revised GPP estimates that exceeded previous values by ~30 %. Much of the knowledge on isotope ratios in atmospheric $CO_2$ has been generated by isotope ratio mass spectrometry (IRMS) measurements in discrete air samples (Ferretti et al., 2000; Werner et al., 2001; Allison and Francey, 2007; Brand et al., 2016), often requiring a complex and well operated logistical network (Ciais et al., 1995a; Welp et al., 2011; Keeling et al., 2017). This may be particularly challenging for tracers such as $\delta^{18}O$-$CO_2$, which can be subject to storage effects in flasks (Rothe et al., 2005; Vardag et al., 2015).

The potential for field-deployable, laser-based instruments measuring both mole fractions and isotope ratios in atmospheric $CO_2$ in real-time has been demonstrated (Bowling et al., 2003; Mohn et al., 2007). Continuous technical improvement of both analysers and applied calibration techniques enables an achievable measurement precision that is increasingly comparable to that of well-performing IRMS systems (Tuzson et al., 2011; Griffith et al., 2012; Sturm et al., 2013; Hammer et al., 2013; Flores et al., 2017; Pieber et al., 2021) and is approaching the compatibility goal of the World Meteorological Organisation (WMO), (Steur et al., 2021). With that, laser-based instruments may be an interesting alternative for observations that were previously limited to IRMS laboratories and flask sampling programmes. The performance of these techniques was demonstrated in the monitoring of annual and seasonal cycles with high temporal resolution (Sturm et al., 2013; Vardag et al., 2016; Pieber et al., 2021). Moreover, real-time measurements of $CO_2$ and its stable isotopes opens new research opportunities that researchers have hitherto not been able to explore. For example, the high temporal resolution achievable with laser-based instruments enables observations ranging from synoptic scales (Vardag et al., 2015; Vardag et al., 2016; Pieber et al., 2021) to that of micrometeorological observations (Griffis et al., 2008; Wehr et al., 2013), for which IRMS based techniques are not a feasible long-term solution.

Commercially available instruments show a large variability in their performance (i.e. achievable measurement precision) and in the level of operator knowledge and interaction required to make meaningful measurements. Key questions are how the achievable instrument performance for each of these instruments compares to traditional IRMS techniques and other laser-based instruments, and what new research opportunities these techniques may provide.



In this study, we assess the performance of an IRIS analyser, the "Delta Ray", manufactured by Thermo (Thermo Fisher Scientific, Bremen, Germany). Note that Thermo has discontinued the manufacturing of the Delta Ray since the experiments of this study were completed. The first studies that deployed the then new Delta Ray instrument in the field have demonstrated its capability to resolve variation ranges in both $CO_2$ mole fractions (101-103 ppm) and isotope ratios (1-15 ‰). These studies included $CO_2$ observations i) at carbon storage and sequestration sites (Van Geldern et al., 2014), ii) of volcanic $CO_2$ emissions (Rizzo et al., 2014; Schipper et al., 2017), iii) at a forest site (Braden-Behrens et al., 2017), and iv) in a cave system (Töchterle et al., 2017). Both Van Geldern et al. (2014) and Braden-Behrens et al. (2017) report Allan deviations for $\delta^{13}C$-$CO_2$ of around 0.04 and 0.02 ‰, respectively. However, both publications also report final measurement uncertainties of their field deployed instruments of 0.3 ‰. Töchterle et al. (2017) and Flores et al. (2017) report uncertainties of 0.34 ‰ and 0.18 ‰ for $\delta^{13}C$-$CO_2$ and 0.44 ‰ and 0.48 ‰ for $\delta^{18}O$-$CO_2$, respectively. Only Rizzo et al. (2014) reports a precision that is close to the values of reported Allan Deviations with <0.05 ‰. Altogether, these studies reveal a significant difference between Allan Deviation and the reported measurement uncertainties, suggesting that the latter may be improved significantly. Achieving uncertainties closer to the compatibility goals of the WMO (WMO-GAW, 2019) potentially increases the variety of research questions this instrument may help to answer.

We tested the Delta Ray instrument in our laboratory and at Baring Head (BHD), our observatory for Southern Ocean background air (Brailsford et al., 2012). Generally, Southern Ocean background air shows very little variability in $CO_2$ mole fractions (Stephens et al., 2013; Steinkamp et al., 2017) and isotope ratios (Allison and Francey, 2007; Moss et al., 2018), providing one of the most challenging environments to explore the performance of a field-deployed instrument for such measurements (Pieber et al., 2021). We compare Delta Ray data with co-located observations of $CO_2$ mole fractions in the context of meteorology and the variability of Radon. Due to the unfortunate failure of the automated air sampler at BHD during the time of the Delta Ray deployment, we cannot provide a direct comparison of the Delta Ray and our well-established GC-IRMS system (Ferretti et al., 2000; Moss et al., 2018). However, we compare the Delta Ray data from background air events with interpolated $\delta^{13}C$-$CO_2$ and $\delta^{18}O$-$CO_2$ observations from BHD and the Cape Grim Observatory (CGO). Despite the limitation to resolve the isotope variability of the smallest $CO_2$ signals (i.e. $CO_2$ variability $\leq 2$ ppm), the Delta Ray time series provides valuable information on the processes controlling the variability of $CO_2$ at BHD on synoptic and daily time scales.

## 2. Methods

We tested the Delta Ray instrument in two separate campaigns. An early Delta Ray model was used during 2015 in NIWA's atmospheric laboratory for Allan Deviation and stability tests, as well as during the deployment at the BHD observatory. In 2018, we used a factory-refurbished Delta Ray model with improved precision. This instrument was used for $CO_2$-free air experiments before it developed faulty behaviour and had to be returned to Thermo without deployment at BHD.



## 2.1 The Delta Ray analyser

The physical principle of the Delta Ray has been described in detail in previous publications (Van Geldern et al., 2014; Töchterle et al., 2017; Braden-Behrens et al., 2017). In short, the instrument measures the absorption spectrum of $CO_2$ in the mid-infrared range. It comprises of two units, the analyser and the Universal Reference Interface (URI). The URI dilutes

isotopically known, pure $CO_2$ gases with $CO_2$-free air and supplies this mixture as reference gas to the laser unit. The operating software (Qtegra) controls the $CO_2$ mole fraction in the reference gas by matching the average mole fraction of a previously measured sample. This concept is applied to account for the $CO_2$ amount effect on the measured isotopic composition of the analyser (Braden-Behrens et al., 2017). The software calculates the isotopic composition of the unknown samples using the isotope values of the two pure $CO_2$ gases. This two-point calibration also accounts for the so-called "scale-compression".

Furthermore, the system requires an air standard with known $CO_2$ mole fractions to calibrate the mole fraction measurements of the Delta Ray. Once all gases are connected and the Delta Ray at operating temperature, the instrument requires a calibration procedure to determine factors for linearity, scale compression, isotope calibration, mole fraction calibration and for the mass flow controller that mixes pure $CO_2$ with the carrier gas. Qtegra applies these factors to all following sample measurements. Thermo specifies the achievable instrument precision to be 0.07 ppm for $CO_2$ mole fractions and as low as 0.05 ‰ for both

$\delta^{13}C$-$CO_2$ and $\delta^{18}O$-$CO_2$ (Thermo, 2014).

## 2.2 Reference gases, quality control gases and carrier gases

Two electropolished 1 L stainless steel flasks were filled to 5 bar with two pure, isotopically distinct $CO_2$ gases, referred to as Marsden and Kapuni. We determined $\delta^{13}C$-$CO_2$ and $\delta^{18}O$-$CO_2$ values of –32.77 ‰ and of –32.52 ‰ for Marsden, and of –

13.75 ‰ and of –11.69 ‰ for Kapuni, respectively (Table 1), using a dual inlet IRMS system and VPDB scale realisation described by Lowe et al. (1994). Over all experiments, we re-filled the flasks with Marsden and Kapuni following the same filling protocol but did not re-calibrate the aliquots. Thereby, we assign a conservative uncertainty of 0.1 ‰ to the isotope values used for both gases (Table 1). While the isotopic difference between Marsden and Kapuni is in the range of the reference gases that Thermo supplies commercially for that purpose ("Ambient" and "Bio"), typical values of atmospheric $CO_2$ are

outside the range covered by our two gases. Due to its isotopic proximity to that of atmospheric $CO_2$, we used Kapuni as the regularly used reference gas (Ref-1) during all measurements, while Marsden was only used during the initial instrument calibration (Ref-2).

We used a $CO_2$-free air from Scott-Marrin (Scott-Marrin, California, USA, now Praxair, USA) as carrier gas in the 2015 campaign. Scott-Marrin produces their $CO_2$-free air by purifying natural air (Scott-Marrin, Lori Thomas, pers. comm, via email

on 16 August 2014) and certifies the $CO_2$-free air with <1 ppm $CO_2$, which exceeded the recommended level of <0.5 ppm (Thermo, 2014). To test the effect of different carrier gases on the measured isotopic compositions, we applied a range of $CO_2$-free gases in the 2018 campaign (section 3).



As calibration gas for $CO_2$ mole fractions, we used a 30 L Luxfer cylinder (Scott-Marrin, California, USA, now Praxair, USA) with compressed natural air, taken at BHD. The $CO_2$ mole fraction was determined by gas chromatography (GC) at NIWA's
atmospheric laboratory.

Furthermore, we used five 30 L Luxfer cylinders with compressed air as quality-control gases (referred to as QC-1 to QC-5). The $\delta^{13}C$-$CO_2$ and $\delta^{18}O$-$CO_2$ isotope ratios of the QC gases were measured on NIWA's gas chromatography isotope ratio mass spectrometer (GC-IRMS) system, (Ferretti et al., 2000; Moss et al., 2018), using a custom made peripheral on a MAT252 isotope ratio mass spectrometer (Thermo Finnigan, Germany). The instrument is calibrated using a propagated VPDB-$CO_2$
scale realisation from the Commonwealth Scientific and Industrial Research Organisation (CSIRO, Aspendale, Australia). Two QC gases were included in each measurement cycle. $CO_2$ mole fractions in QC-1 to QC-3 were determined using a gas chromatograph system at NIWA's gaslab, while mole fractions of QC-4 and QC-5 were measured using the Picarro system at BHD and via comparison of peak sizes on the GC-IRMS instrument, respectively. $CO_2$ mole fractions in QC-1 to QC-4 are calibrated to the WMO $CO_2$ X2007 scale.


## 2.3 Calibration scheme

Thermo designed an integrated referencing technique for the Delta Ray, in which two pure $CO_2$ gases with known isotopic composition get diluted with $CO_2$-free air to match $CO_2$ mole fraction range of measured air samples. Therein, the system attempts to conserve the isotopic composition of the pure $CO_2$ during the dilution process and to provide a $CO_2$-in-air reference
gas to the analyser that has constant isotope ratios at dynamic $CO_2$ mole fractions. The main purpose for this technique is to account for non-linear, $CO_2$ mole fraction dependent isotope effects (Thermo, 2014).

However, the application of this compulsory referencing technique does not follow the Principle of Identical Treatment (Werner and Brand, 2001), hereafter referred to as PIT, which is regarded as the "golden rule" for isotope referencing in the IRMS community. Therefore, we treat the data output of the Delta Ray as preliminary. We designed our measurement
sequences to include two quality control gases (QC-1, QC-2, QC-3) and used QC-1 as working standard to reference the Delta Ray data to the VPDB isotope scale (Table 1), which followed the PIT and improved the reproducibility of QC-2 and QC-3 significantly.

## 2.4 Configuration of analytical setup and measurement sequence

The Delta Ray system configuration comprised of the Delta Ray analyser, two pure $CO_2$ gases (Kapuni and Marsden), one $CO_2$-free air as carrier gas, one air standard for $CO_2$ mole fractions and two QC gases at a time (Fig. 1). Kapuni and Marsden are configured in the Qtegra as Reference-1 and Reference-2 (Ref-1 and Ref-2), respectively. The Delta Ray analyser includes an inbuilt Nafion system (Nafion, Permapure, USA). However, because this inbuilt Nafion combines the air intake and air outlet in counterflow without water removal, its efficacy is not clear to us, as it potentially mediates water in both directions.





Therefore, we used an additional Nafion membrane to dry the incoming air, where the outgoing air was dried using a molesieve
trap before it was used as drying air in the Nafion membrane. Three-way solenoid valves were used to switch between air
samples and QC gases. The valves were configured so that air samples were measured in the "normally closed" position, to
ensure the QC gas cylinders were closed during potential electric failure. Air samples and QC gases were introduced via
"sample port B", to fulfil the PIT as much as possible.


The Delta Ray has the capability to control up to four external solenoid valves and we used two of these to switch between
ambient air and the two QC gases. Measurement sequences were defined in Qtegra and executed in continuous cycles (Table
2). Each measurement sequence begins with the measurement of Ref-1, which is followed by measurement blocks of QC-1,
air sample and QC-2, before another Ref-1 measurement block marks both the end of the current measurement sequence and
the start of the next measurement sequence. We allowed a flush time of 150 s after each gas change, to ensure complete gas
replacement in the optical cell. Thereafter, Ref-1 was measured for 300 s, while air and QC gases were measured in three
blocks of 200 s each, leading to a total time of 45 minutes per measurement sequence. This measurement sequence resulted in
only 10 min of effective air measurement in every 45 minutes. Because the objective of this test was to assess the analyser
performance, our sequence included a disproportional amount of QC gas measurements. For the data analysis, we calculated
average values of each measurement block, resulting in ~32 data points for air and QC gases per 24 h period.

Considering the gas flow-rate of 80 mL min$^{-1}$ through the analyser, as well as the measurement and flush times we defined for
reference- and QC gases in the measurement sequence, the Delta Ray consumed 19 L of $CO_2$-free air and 32 L of each of the
two QC gases per day. In this configuration, cylinders with 30 L volume filled to 138 bar (2000 PSI) would be exhausted in
215 days for $CO_2$-free air and in 129 days for QC gases. This would require two to three cylinder replacements per year (Table
2). The QC gas consumption would decrease significantly with greater proportion of air measurements. However, Thermo
recommends a frequency of Ref-1 measurements of 1 per 30 minutes, which means the $CO_2$-free air consumption should not
be decreased further.

**2.5 Data correction and uncertainty propagation with QC-1**

QC-1 was selected as working standard to convert all mole fraction and isotope ratio measurements to the respective scales
(Table 1). QC-1 comprised of natural air with similar mole fractions and isotope ratios of $CO_2$ to the air measured during the
campaign to fulfil the PIT.

We derived unprocessed values for $\delta^{13}C$-$CO_2$, $\delta^{18}O$-$CO_2$ and $CO_2$ in QC-1 of $-8.74 \pm 0.06$ ‰, $-1.22 \pm 0.07$ ‰ and $400.88 \pm$
$0.19$ ppm, respectively. The comparison to the calibrated values (Table 1) suggests that the long-term averages of the Delta
Ray measurements in QC-1 are too depleted in both $^{13}C$ and $^{18}O$ by $-0.20 \pm 0.13$ ‰ and $-0.60 \pm 0.11$ ‰, respectively, while
they are too high in $CO_2$ by $0.45 \pm 0.28$ ppm (Table 3). The magnitude of this offset is consistent for QC-2 and QC-3, suggesting





a correction for the offset in QC-1 is suitable for all measured parameters in all gases. We correct all Delta Ray measurements
according to:


$$X_{s(n)} = X_{DR-s(n)} - \left(X_{DR-QC1(n)} - X_{scale-QC1}\right) \tag{1}$$

where $X_{s(n)}$ is the calibrated sample average as measured in sequence n on the respective scale, $X_{DR-s(n)}$ and $X_{DR-QC1(n)}$ are the
measurement averages for the sample and QC-1 within measurements sequence n, while $X_{scale-QC1}$ refers to the calibrated value
for QC-1 (Table 1).

Likewise, we calculated the uncertainty of $X_{s(n)}$ as


$$U_{s(n)} = \sqrt{U_{DR-s(n)}^2 + U_{DR-QC1(n)}^2 + U_{scale-QC1}^2} \tag{2}$$

where $U_{s(n)}$ is the fully propagated uncertainty of each sample in measurement sequence n, while $U_{DR-s(n)}$ and $U_{DR-QC1(n)}$ are the
standard deviations of the Delta Ray measurements for sample and QC-1 in measurement sequence n, respectively (Section
2.4). $U_{scale-QC1}$ refers to the uncertainty of the target value assignment of QC-1 (Table 1). This correction and uncertainty
propagation is applied to all $\delta^{13}C$-$CO_2$, $\delta^{18}O$-$CO_2$ and $CO_2$ measurements in air, QC-2 and QC-3.

### 3. Assessing instrument sensitivity to variable qualities of $CO_2$-free air as carrier gas

#### 3.1 Quality requirements for $CO_2$-free air as carrier gas

Optical analysers for measurements of air samples, such as the Delta Ray, are sensitive to changes in the composition of the
air matrix, i.e. the mole fractions of $N_2$, $O_2$ and Ar (Werle et al., 1993; Chen et al., 2010; Thermo, 2014), referred to as the
pressure-broadening effect. To prevent the pressure-broadening effect, it is of paramount importance that the air matrix in air
samples and reference gases is identical, following the PIT. For measurements of natural air samples with the Delta Ray, the
$CO_2$-free air used as carrier gas must therefore comprise of a natural, ultra-pure air matrix ($N_2 = 78$ %, $O_2 = 21$ %, Ar $= 1$ %)
and have a $CO_2$-blank of <0.5 ppm (Thermo, 2014).
The calibration strategy of the Delta Ray setup as recommended by Thermo is largely dependent on the quality of the $CO_2$-
free air, where $CO_2$-free air of sub-optimal quality may limit the achievable accuracy of the system. Because of that, Delta Ray
users need to manage their long-term $CO_2$-free air requirements in addition to their reference gas usage. For example, research
applications such as long-term atmospheric monitoring with measurement focus on very small signals also require the lowest





possible variability in both air matrixes and $CO_2$ blanks between consecutive $CO_2$-free air cylinders. The setup we built for

this study uses Ref-1 as "mediator" only. The final referencing of air measurements is based on QC-1, which fulfils the PIT and therefore mitigates potential variability due to variation in the quality of consecutive $CO_2$-free air supply.

## 3.2 $CO_2$-free air mixed from pure $N_2$ and $O_2$

Commercial $CO_2$-free air can be manufactured by mixing main air components of high purity, such as the "Ultra-zero grade

air" from BOC (BOC, Linde Group, Wellington, New Zealand) with $CO_2 \leq 1$ ppm, $O_2 = 21 \pm 1$ %, $Ar = 0$ %, in $N_2$ balance. Ultra-zero grade air is the highest quality zero air product that is readily available in New Zealand, however, its certified values neither satisfy the quality criteria for the remaining $CO_2$ level, nor for the composition of the matrix air. This highlights potential logistical limitations to source suitable $CO_2$-free air from local gas providers. We attempt to quantify the consequences that can be expected when the $CO_2$-free air is not meeting requirements. Because instruments in our laboratory

are not calibrated for sub-ppm measurements of $CO_2$ mole fractions, we introduce the $CO_2$-free gases into the Delta Ray and measure the transmission in the absorption spectra to get a quantitative estimate of the $CO_2$ blank (Fig. 2). With a certified $CO_2$ blank of $\leq 1$ ppm, the Ultra-zero grade air from BOC resulted in a transmission of 0.7 % on the main peak of the main isotopologue (indicated as 626 for $^{16}O^{12}C^{16}O$). Passing the Ultra-zero grade air through a chemical $CO_2$ scrubber (Carbosorb, Elemental Microanalysis, Devon, UK) reduced the transmission to 0.2 %. By comparing the two transmission values, we think

that it is likely that the chemical scrubbing reduced the $CO_2$ blank to $\leq 0.5$ ppm and thus meets the manufacturer's $CO_2$-blank requirements. Further $CO_2$-removal from commercial carrier gases may be required to achieve acceptable $CO_2$ levels.

## 3.3 $CO_2$-free air from purified natural air

$CO_2$-free air can also be prepared by removing $CO_2$ from natural air, which minimises the potential to alter the composition of

the air matrix. For the 2015 campaign, we sourced "ultrapure" air from Scott Marrin (now Praxair, Linde Group, Pennsylvania, USA), which is made from purified natural air (Scott-Marrin, Lori Thomas, pers. comm, via email on 16 August 2014). In the 2018 campaign, we prepared $CO_2$-free air in NIWA's atmospheric laboratory. Therefore, we use an oil-free compressor (Sweetair, SA-6E, RIX, California, USA) with a 13X molesieve trap (8-12 mesh Sigma-Aldrich) on the compressor inlet in combination with a chemical $CO_2$ scrubber (Carbosorb, Elemental Microanalysis, Devon, UK) on the compressor outlet, and

filled a 30 L cylinder to 50 bar.

The $CO_2$-free air produced at NIWA showed a transmission of 0.2 % (Fig. 2). Based on the experiments with the Ultra-zero grade air from BOC, we estimate the $CO_2$-blank in the $CO_2$-free air produced at NIWA to be $\leq 0.5$ ppm as well. Because the added Carbosorb trap in the experiments with the Ultra-zero grade air from BOC produces $CO_2$ blanks that are indistinguishable from the $CO_2$ blank in the $CO_2$-free air made at NIWA, we conclude that adding the Carbosorb trap not only





minimises the $CO_2$ blank as much as possible but it also homogenises the $CO_2$ blanks between different $CO_2$-free air cylinders, which would minimise long-term variability.

Measurements of the $O_2/N_2$ ratio in the $CO_2$-free air prepared at NIWA confirmed the natural composition of the air matrix was preserved during the purification step. Because of that, we assume that natural $Ar/N_2$ ratios were preserved as well and that atmospheric measurements referenced with purified natural air as $CO_2$-free air do not create an accuracy offset due to

pressure broadening. In comparison to the Ultra-zero grade air from BOC with an uncertainty of the $O_2$ mole fraction of $\pm$ 1 %, $CO_2$-free air produced with this technique also guarantees a minimal variability in the air matrix between different $CO_2$-free air cylinders and hence minimal accuracy offsets in long-term measurement series.

**3.4 Accuracy offsets due to pressure broadening effects in $CO_2$-free air**

To assess the effect of the different $CO_2$-free carrier gases on the isotope measurements, we measured the two cylinders, QC-4 and QC-5 (Table 1) on the Delta Ray setup, using Ref-1 and different carrier gases: i) $CO_2$-free air prepared at NIWA, ii) Ultra-zero grade air from BOC, iii) Ultra-zero grade air from BOC with a Carbosorb trap to reduce the $CO_2$-blank. We found systematic variation in the measured values. While the measurements with the $CO_2$-free air prepared at NIWA produced accurate isotope values in QC-4 and QC-5 within 0.2 ‰, measurements made with the Ultra-zero grade air from BOC resulted

in offsets in the range of $+1.14 \pm 0.11$ ‰ for $\delta^{13}C$-$CO_2$ and of $+0.15 \pm 0.04$ ‰ for $\delta^{18}O$-$CO_2$. Reducing the $CO_2$ blank in the Ultra-zero grade air from BOC resulted in slightly larger offsets of $+1.24 \pm 0.13$ ‰ for $\delta^{13}C$-$CO_2$ and in comparable offsets of $+0.17 \pm 0.03$ ‰ for $\delta^{18}O$-$CO_2$. This suggested that the air matrix effect was dominating the offset and that the Ultra-zero grade air from BOC was not a suitable $CO_2$-free air for the Delta Ray system. Therefore, we operated the Delta Ray with purified natural air as carrier gas.


**4. Assessing instrument performance using QC gas measurements in the laboratory**

**4.1 Allan deviation**

We determined the Allan deviation of an earlier version of the Delta Ray analyser during the 2015 campaign using QC-1 and find values of 0.03 ‰ for both $\delta^{13}C$-$CO_2$ and $\delta^{18}O$-$CO_2$ and <0.01 ppm for $CO_2$ for integration times of 200 to 300 s (Fig. 3).

These values are comparable to the findings of Braden-Behrens et al., (2017). We use our Allan Deviation results in the design of our measurement sequence and schedule blocks between 200 and 300 s for air and QC gases (section 2.4).



### 4.2 Instrument stability during six hours of QC gas measurements

To test the instrument stability of the Delta Ray in the laboratory prior to deployment at BHD, we measured sequences of QC-1 and QC-2 over 6 h. After 5 h, we observed a sudden 0.4 ‰ shift in both $\delta^{13}C$-$CO_2$ and $\delta^{18}O$-$CO_2$ traces that occurred simultaneously for both gases, while the mole fraction measurements of both gases remained unaffected (Fig. 4). We have no explanation for the sudden shifts at this point but can think of two potential causes for this artefact: i) The Delta Ray experiments were performed in a laboratory of which the temperature was not tightly controlled. Because the laboratory had no external walls or windows, temperature fluctuations were likely below $0.2°C$ $min^{-1}$, which Thermo specifies as acceptable temperature gradients. While we cannot rule out that a greater temperature change occurred, we think it is unlikely that the temperature in the laboratory suddenly changed dramatically. The experiment was made during the period of core working hours, when both the traffic in and out of the laboratory as well as the magnitude of traffic-induced temperature changes are at their maximum. If the laboratory temperature was not suitable for the Delta Ray, we would expect similar shifts during the first 5 h of this experiment, which we did not observe. ii) We speculate that instabilities during the referencing step may have caused that artefact, which would create a simultaneous shift in both QC gases of identical magnitude. We think that this is the most likely explanation, however, we have no means to support this hypothesis. Interestingly, we notice a significant variability in the $CO_2$ mole fraction measurements in Figure 7 of Braden-Behrens et al., (2017) during measurements of their "target gas", which is not reflected in their isotope traces. Furthermore, Figure 7 in Braden-Behrens et al., (2017) shows the same pattern of sudden, synchronous changes in the $\delta^{13}C$-$CO_2$ and $\delta^{18}O$-$CO_2$ measurements of the "target gas" that we observe and describe here. To monitor such artefacts in the following experiments and to be able to correct for such effects, we measured two independent QC gases in every measurement sequence and to apply a calibration scheme that is not entirely based on Ref-1. This enables unambiguous identification whether instability originates from the Delta Ray instrument or the reference gases and it provides the means to remove affected data, or to correct for such effects.

### 5. Instrument deployment at BHD, site description and typical air advection pattern observed during instrument deployment

#### 5.1 The BHD site

The Delta Ray was deployed at BHD, located on the edge of an 85 m southward-facing cliff, overlooking the Southern Ocean (41.4083°S, 174.8710°E). BHD lies within a regional park at the southern tip of the greater Wellington region with a population of 520,000. Atmospheric dynamics at BHD are highly variable and complex but show a distinct pattern that is described in detail by Brailsford et al., (2012) and Steinkamp et al., (2017). The topography of New Zealand's North and South Islands deflect the flow path of advected air masses so that the resulting wind direction at BHD ranges between either north-west to north, or south-west to south-east most of the time (Fig. 11).



When the air is advected from between north-west to north, the air has most likely passed over New Zealand's North Island
and potentially includes a significant terrestrial signal. In some cases, air arrives at BHD from true north but has been deflected
from further west to south-west where it has passed over the Tasman Sea and does therefore not carry a clear terrestrial signal
at all. Furthermore, it may have passed over the northern parts and the West Coast of New Zealand's South Island, in which
case it potentially includes a terrestrial signal. Similarly, air advected from south-west to south has potentially passed over the
East Coast of New Zealand's South Island, a region marked by major cities and agricultural activity. In summary, air masses
advected from either north or south may include a terrestrial signal but may also represent oceanic air. Only air advected from
between south to south-east has originated from the Southern Ocean and has not been in contact with land masses for many
days. These air masses are amongst the cleanest on the planet and are representative of baseline air. During baseline conditions,
the variability of $CO_2$ mole fractions can be less than 0.1 ppm over several hours and even days (Brailsford et al., 2012;
Stephens et al., 2013; Steinkamp et al., 2017). In contrast, northerly air masses that contain a terrestrial $CO_2$ signal are likely
to include anthropogenic $CO_2$ emissions from urban areas in the greater Wellington region as well. During periods of low wind
speeds, the measured $CO_2$ signal can be dominated by local biogeochemical $CO_2$ fluxes, where the short-term variability in
$CO_2$ mole fractions can exceed 10 ppm (Stephens et al., 2013; Steinkamp et al., 2017). However, the short-term variability in
$CO_2$ isotope ratios has not yet been quantified during such conditions at BHD.

Flask samples are routinely taken during baseline events and include the analysis of $CO_2$ isotope ratios at NIWA's atmospheric
laboratory in Wellington (Ferretti et al., 2000), as well as for intercomparison programmes with the Scripps Institute of
Oceanography and the Institute for Alpine and Arctic Research (Moss et al., 2018). The $\delta^{13}C$-$CO_2$ time series from BHD shows
a variability that is typically within 0.2 ‰ per year and a long-term trend of ~0.3 ‰ per decade towards $^{13}C$ depletion, largely
a result from continuously added $CO_2$ from fossil fuel combustion, referred to as the Suess-effect (Keeling et al., 2017).
The BHD observatory is home to many different analytical systems. Continuous measurements of $CO_2$ mole fractions have
been performed at BHD since 1972. A Siemens Ultramat 3 analyser was used from 1985-2016 (Brailsford et al., 2012), while
a Picarro G2301 analyser was installed in 2011 (Steinkamp et al., 2017). A Radon analyser was installed in 2015 by the
Australian Nuclear Science and Technology Organisation (ANSTO), providing half-hourly average data that indicate the
degree to which the measured air mass has been in contact with land masses before reaching BHD (Williams et al., 2011;
Chambers et al., 2016). The tower at BHD is equipped with a range of meteorological sensors at 12 m above ground level (Fig.
5). Wind data are measured by 2-D ultrasonic anemometer (Wind Observer II, Gill Instruments, UK) which was installed in
May 2013. The raw wind components are measured at 2 Hz and converted to a 3 s average. The 3 s vector components are
averaged to 10 minute and hourly wind statistics and stored with other meteorological variables in the station data logger
(CR1000, Campbell Scientific Inc, USA). Wind characteristics of the site are described by Stephens et al., (2013). The
temperature in the laboratory where the Delta Ray was operated was controlled to $19.5 \pm 1.5°C$, while larger temperature
changes may occur during weekly maintenance visits.





**5.2 Wind direction, wind speed and Radon variability during Delta Ray deployment at BHD**

Because the variability of $CO_2$ mole fractions measured at BHD is strongly controlled by atmospheric advection (Brailsford et al., 2012; Stephens et al., 2013; Steinkamp et al., 2017), we expected that this also applies to the isotopes of $CO_2$. Therefore,

we will briefly describe characteristics of selected advection patterns that were observed during the Delta Ray deployment at BHD. We support the interpretation of our meteorological observations from BHD with back trajectories from HYSPLIT, an atmospheric transport and dispersion model (Stein et al., 2015; Rolph et al., 2017). Figure 11 shows 30 min averages of Radon data, and hourly averages of both wind speed and wind direction from BHD. Wind direction data are clustered into eight sectors of 45°, i.e. with the centres of clusters north and south being 360/0° and 180°, respectively.


We observed three significant southerly events resulting in baseline/or near baseline $CO_2$ values on 27 May, 5 June, and 15 June of 2015 (S1, S2 and S3 in Figure 11). These three southerlies are marked by some of the lowest Radon levels in the record, suggesting the measured air had no significant contact with land masses during the days before advection to BHD. While S2 fulfils the strict requirements for a baseline-air event (Brailsford et al., 2012; Steinkamp et al., 2017) between 08:30

and 18:30 on 5 June 2015, both S1 and S3 are not classified as baseline-air events. The accuracy of this classification is corroborated by HYSPLIT back-trajectories for S1 and S3, showing that the air has travelled over New Zealand's South Island before it was measured at BHD, suggesting that air from both S1 and S3 may contain a terrestrial component. In contrast, back trajectories show that air masses measured during S2 have been advected from the Southern Ocean, without direct contact to New Zealand's South Island (Fig. 13).


Furthermore, we observe two long-lasting northerly events from 29 May to 1 June 2015, and from 8 to 11 June 2015 (N1 and N2 in Figure 11). While the average hourly wind speed of N1 with ~10 m s$^{-1}$ seems typical for our study period, average wind speeds persistently exceeded ~20 m s$^{-1}$ during N2. The very low Radon levels during N2 are comparable with those from S1-S3, indicating the terrestrial impact on the measured $CO_2$ during N2 was small. Back trajectories for the N2 event show that

the air was indeed advected from the Ocean, with very limited contact to land masses before reaching BHD (Fig. 12, F).
We find that the deployment at BHD provides an analytically challenging environment for the Delta Ray analyser, enabling the assessment of its capability to resolve very small to moderate changes in $CO_2$ mole fractions and isotope ratios. Particularly the three southerlies (S1-S3) with very similar properties provide an opportunity to assess the performance of the Delta Ray system under field conditions at a baseline observatory.



## 6. Assessing instrument performance during deployment at BHD

### 6.1 Outlier detection in BHD time series using QC gases

Figure 6 shows all measurements of QC gases during the deployment at BHD as 10-minute averages (n=791). All values in this figure are shown as provided by the Delta Ray, i.e. as measured against Ref-1, without further data processing. On 11 June 2015, the QC-2 was nearing a very low pressure and was replaced with QC-3.

The $\delta^{13}C$-$CO_2$ and $\delta^{18}O$-$CO_2$ time series from the QC gases vary around their long-term average and do not show a long-term drift (Fig. 6). However, periods of strong variability appear on 21 May, on 25 May and from 10 to 11 June of 2015 that impact on all isotope measurements within the respective 45 min measurement cycle. We did not find a reason for the increased variability. We think that we can rule out that sudden temperature changes have caused the sudden variability, because the most abrupt temperature changes this time of year occur when the door is open during maintenance visits. However, maintenance visits didn't coincide with the periods of increased variability in the record.

We find that $\delta^{13}C$-$CO_2$ and $\delta^{18}O$-$CO_2$ are affected in the same order of magnitude and that both isotope traces are affected simultaneously while the $CO_2$ mole fraction data are not affected at all. This pattern is identical to the sudden shift that we observed in earlier laboratory tests (Section 4.2) and is similar to observations of Braden-Behrens et al., (2017). Measurement sequences are flagged as outlier (yellow symbols in Figure 6), when the $\delta^{13}C$-$CO_2$, the $\delta^{18}O$-$CO_2$ or the $CO_2$ measurements of QC-1 deviate from their long-term average by more than three standard deviations (3 σ). We reject 20 measurement sequences, affecting around 2.5 % of the measurements, resulting in a total of 791 measurement sequences from the 26 day deployment period. For an unknown reason, the measurements of both isotope ratios in QC-2 show a systematic variability, which does not occur in QC-1 and QC-3 (Fig. 6 and 8), resulting in generally larger standard deviations of the QC-2 data (Table 3).

### 6.2 Reproducibility of QC gas measurements during deployment at BHD

Following the removal of outliers, we use the standard deviation (1 σ) of the three QC gas measurements within each measurement cycle as indicator for the reproducibility of the Delta Ray setup. This provides us with 791 values for QC-1, 661 for QC-2 and 130 for QC-3. The histograms in Figure 7 display the distribution of the standard deviation values, showing that the majority of the $\delta^{13}C$-$CO_2$ and $\delta^{18}O$-$CO_2$ values lie within a range that is comparable to the Allan Deviation of 0.03 ‰ (Section 4.1 and Fig. 3). This suggests that the Delta Ray system at BHD is operating close to its maximum performance level for isotope ratio measurements during most of the time. In contrast, the histogram for $CO_2$ shows that all standard deviation values from all three QC gases are out of range compared to the Allan Deviation of 0.01 ppm (Section 4.1). Furthermore, the standard deviations of the $CO_2$ measurements appear in QC gas specific clusters. The reason for this pattern and the weak performance in $CO_2$ measurements remains unclear but is likely associated with effects in the cylinders or pressure regulators rather than the Delta Ray itself.



## 6.3 Control of linearity and isotope scale compression during deployment at BHD

We compare the average measurement results for the QC gases obtained by the Delta Ray measurements to the target values determined by GC and GC-IRMS analysis in Table 3. QC-1 and QC-2 were designed to cover a large range in $CO_2$ mole
fractions (~97.3 ppm) as well as in $\delta^{13}C$-$CO_2$ (~4.9 ‰) and $\delta^{18}O$-$CO_2$ (~5.5 ‰), to assess the capability of the Delta Ray to make accurate measurements over a large range. Table 3 shows good agreement between the Delta Ray measurements and the target values. This suggests that a potential linearity effect is sufficiently controlled via the linearity calibration of the Delta Ray, and that the calibration scheme of the Delta Ray based on Ref-1 and Ref-2 is able to prevent significant scale compression artefacts.


However, the QC-1 corrected data seem to overestimate the $CO_2$ mole fraction in QC-2 by about 0.27 ppm, given a $CO_2$ difference between QC-1 and QC-2 of 97.3 ppm. In comparison, this difference accounts for only 0.01 ppm for QC-3, which had $CO_2$ mole fractions that were very similar to that of QC-1. While this difference is within the combined measurement uncertainty for $CO_2$ mole fractions in both cases, it might be due to inaccurate control of large $CO_2$ variations. If this
overestimation in QC-2 was based on a linear process, it would add an error of $+0.0028$ ppm ppm$^{-1}$ to the mole fraction measurements. Using this value, we estimate that the $CO_2$ mole fractions in the air measurements of the deployment at BHD would need to exceed or fall below the target value of QC-1 ($400.43 \pm 0.09$ ppm) by >18 ppm to produce an offset that exceeds the compatibility goal formulated by the WMO, which did not occur during the deployment at BHD. Note that the Delta Ray has the capability for a two-point mole fraction calibration, which we have not utilised during this assessment. It is thus likely
that the control of this effect can be improved by a second "concentration standard" (Thermo, 2014).

## 6.4 $CO_2$ mole fraction measurements in QC gases

The $CO_2$ mole fraction data from the QC gas cylinders show synchronous variations of similar magnitude (Fig. 6). Interestingly, this feature is similar to observations made by Braden-Behrens et al., (2017), who also found a similar variability
in $CO_2$ mole fraction measurements in cylinder air. A linear regression analysis between QC-1 and QC-2 suggests that about 84 % of the variability in the mole fraction measurements in both cylinders can be explained by the same process. This finding gives strong support for using QC-1 as the working standard in the post-processing protocol for $CO_2$ mole fractions. Moreover, this highlights the importance to determine and apply correction factors for every single measurement sequence. It is important to note that the prominent features in the $CO_2$ mole fraction measurements are not reflected in the isotope traces. This suggests
that the internal linearity calibration of the Delta Ray is robust for $CO_2$ variations of that magnitude (Fig. 6).

**6.5 Assessment of the instrument performance using QC gas measurements from BHD in performance chart method**

Given the lack of a second QC gas that was measured over the entire campaign and the low quality of the measurements of QC-2, we use the QC-3 time series to assess the reproducibility of the Delta Ray measurements using the performance chart

method (Werner and Brand, 2001) in Figure 9. The performance chart is based on $\delta^{13}C$-$CO_2$, $\delta^{18}O$-$CO_2$ and $CO_2$ values of QC-3, after full corrections have been applied. Error bars represent the fully propagated uncertainty of the measured averages in each sequence. Next, we determine the standard deviation (1 $\sigma$) of all $\delta^{13}C$-$CO_2$, $\delta^{18}O$-$CO_2$ and $CO_2$ values from all QC-3 measurements as indicator of the instrument performance (Werner and Brand, 2001). We find a reproducibility (1 $\sigma$) for $\delta^{13}C$-$CO_2$, $\delta^{18}O$-$CO_2$ and $CO_2$ of 0.07 ‰, 0.06 ‰ and 0.03 ppm, respectively, n = 130, which we use as a measure of achievable

measurement precision. Because the variability in isotope ratios of the relatively short time series for QC-3 is similar to the time series of QC-1 spanning 26 days (Fig. 6, 7 and 8), we think this is a representative estimate.

While the precision estimates for both isotope ratios did not meet the WMO network compatibility goal of 0.01 ‰ for $\delta^{13}C$-$CO_2$ and 0.05 ‰ for $\delta^{18}O$-$CO_2$, they did meet the expanded compatibility goal of 0.1 ‰ for both parameters. However, our

instrument precision for $CO_2$ mole fractions of 0.03 ppm met the WMO network compatibility goal of 0.05 ppm (WMO-GAW, 2019).

**7. Assessing the instrument performance by analysing 26 day time series from deployment at BHD**

The following sections compare the Delta Ray time series to observations made with well-established measurement systems

at BHD. Furthermore, we describe and interpret features in the Delta Ray time series in the context of atmospheric advection, with the objective to highlight the capability of the Delta Ray instrument to resolve the variability of $CO_2$ and its isotope ratios at BHD under field conditions.

**7.1 Comparing $CO_2$ mole fraction measurements from Delta Ray and Siemens Ultramat 3 at BHD**

We used 5 min average $CO_2$ mole fraction measurements from the Siemens Ultramat 3 gas analyser at BHD (Brailsford et al., 2012; Stephens et al., 2013). To compare Siemens and Delta Ray data, we removed periods when the Siemens was in calibration mode from both time series. Next, we sub-sampled the remaining 5 min averages from the Siemens at the time averages of the remaining Delta Ray data by linear interpolation and used the resulting 738 data pairs for comparison.

Figure 10 displays the $CO_2$ mole fraction data comparison between the Delta Ray and the Siemens analysers. The histogram showed the residuals with a Gaussian distribution, suggesting that the offset was not systematically biased towards either lower





or higher mole fractions. The potential scale effect in our Delta Ray setup of 0.0028 ppm ppm$^{-1}$ produced a maximum bias of 0.04 ppm on the 29 May 2015 (Section 6.3). Interestingly, the slope of 0.97 in the comparison confirmed a bias in the Delta Ray data towards higher values (Fig. 10). Out of the 738 data pairs used for the comparisons, 155 or 21 % agreed within the

WMO compatibility goal of 0.05 ppm for the Southern Hemisphere, while 287 or 39 % of the data pairs agreed within the WMO compatibility goal of 0.1 ppm for the Northern Hemisphere, respectively (WMO-GAW, 2019). Furthermore, 542 or 73 % of the data pairs agreed within the standard deviation of the mole fraction averages, which amounted up to a few ppm during times of high $CO_2$ variability in the measured air (Fig. 10). In these cases, the large standard deviation coincided with large residual values, suggesting that the agreement between both time series could be improved through synchronisation of

the averaging intervals.

This finding is corroborated by the excellent reproducibility of $CO_2$ mole fraction measurements of QC-3 in the performance chart of 0.03 ppm (Section 6.5), suggesting the Delta Ray setup is capable of highly reproducible mole fraction measurements that meet the WMO network compatibility goal.

**7.2 $CO_2$ mole fraction observations of the Delta Ray during deployment at BHD**

The $CO_2$ mole fractions during the Delta Ray deployment at BHD varied between 392 ppm and 414 ppm, spanning a total range of 22 ppm. The average $CO_2$ mole fraction of 397.09 ppm from the baseline event S2 that occurred on 5 June 2015 is shown with a red line in Figure 11, C.

The most prominent pattern of the $CO_2$ mole fraction time series is the daily cycle, which is typically characterised by $CO_2$

minima between midday and later afternoon when photosynthetic $CO_2$ uptake dominates $CO_2$ fluxes; and $CO_2$ maxima at night-time during boundary layer build-up of $CO_2$ from respiration and anthropogenic sources. $CO_2$ peaks are typically accompanied by Radon peaks of proportional magnitude (Fig. 11, D), highlighting the interplay of $CO_2$ dilution by wind speed versus $CO_2$ accumulation in the boundary layer, as control of the $CO_2$ peak amplitude (Williams et al., 2011; Chambers et al., 2016).

The largest single event of the $CO_2$ time series is the $CO_2$ build-up to 414 ppm in the night-time boundary layer in the early hours of 29 May 2015, also coinciding with the largest peak in the Radon time series (Fig. 11, C and D). Back trajectories show the air flow leading up to this event (Fig. 12, B). In the 48 h before measurement at BHD, the measured air had passed over the cities of Dunedin, Christchurch, Lower Hutt and Wellington in a southerly, before the wind direction changed to northerly and the same air passed over Wellington and Lower Hutt again before measurement at BHD. The amplitude of the

$CO_2$ peak was enhanced by the relatively low wind speeds of $<5$ m s$^{-1}$, preventing effective vertical mixing. The advection pattern suggests that both urban $CO_2$ emissions and ecosystem respiration contributed to the elevated $CO_2$ levels.

We observe seven daily $CO_2$ cycles with an amplitude between 10-15 ppm occurring after 1 June 2015. These events are typically associated with air advection across New Zealand's North Island (Fig. 12, E) and moderate wind speeds. However,





only four of these events coincide with high Radon peaks (Fig. 11, D). The data gap in the Radon time series on 1 June 2015 is because the instrument is in calibration mode on the first day of every month. Furthermore, nine daily $CO_2$ cycles with amplitudes between 5-10 ppm and three with amplitudes between 1-5 ppm occur in the record, where most of them have a corresponding signal in the Radon time series. However, some exceptions occur where days with ~6 ppm (14 June 2015) or even ~13 ppm (2 and 3 June 2015) amplitudes of the daily $CO_2$ cycle don't show a significant counterpart in the Radon time series.

Both the amplitude and the timing of these twenty daily $CO_2$ cycles are controlled by wind direction and wind speed. In general, the amplitude of the daily $CO_2$ mole fraction cycle appears in an inverse relationship with wind speed in the measurements of terrestrial air, where higher wind speeds coincide with smaller amplitudes and peak widths in the daily $CO_2$ cycle. For example, the largest $CO_2$ peak during 29 May 2015 occurred when wind speeds were below 5 m s$^{-1}$. In contrast, the daily $CO_2$ cycle was strongly dampened to values between 2 and 4 ppm at persistent wind speeds of ~10 m s$^{-1}$ between 30 to 31 May 2015 (N1), when the air was advected along the west coast of the North Island (Fig. 12, C). We observed $CO_2$ cycles with an even smaller amplitude of ~1 ppm between 9 and 10 June 2015 (N2). Interestingly, $CO_2$ minima and maxima appear out of phase with the expected timing of daily $CO_2$ cycle during these days. While the recorded wind direction is true north, back-trajectories show that the air has been deflected and in fact originated from the Tasman Sea, suggesting that the small $CO_2$ variability could be a distant signal (Fig. 12, F).

As expected for air with Oceanic properties, we observe no daily $CO_2$ cycle during S1-S3. With the onset of a southerly event, an ongoing daily cycle diminishes, and $CO_2$ mole fractions begin to approach baseline values (Fig. 11, C). Because the air during a southerly event becomes more stable and cleaner with duration of the event, we focus on the final 6 h period within each S1, S2 and S3. We found similar $CO_2$ mole fractions for S1 and S3 of 397.29 ± 0.07 ppm and 397.21 ± 0.05 ppm, respectively, while the baseline air event S2 is marked by slightly lower $CO_2$ mole fractions of 397.09 ± 0.11 ppm. HYSPLIT back trajectories show the S2 has not been in contact with land masses prior to the measurements, while air masses measured during both S1 and S3 have been in contact with New Zealand's South Island (Fig. 13). The small difference between S2 on the one hand and both S1 and S3 may thus be a result from an additional component of terrestrial $CO_2$ during S1 and S3.

### 7.3 $\delta^{13}$C-$CO_2$ observations of the Delta Ray during deployment at BHD

The $\delta^{13}$C-$CO_2$ data from the field deployment at BHD appear with an average value of about –8.5 ‰. The most prominent features in the $\delta^{13}$C-$CO_2$ time series are the systematic daily cycles that occur in concert with $CO_2$ mole fractions (Fig. 11, B). The $\delta^{13}$C-$CO_2$ maxima in observed daily cycles are marked with C and are numbered in the $\delta^{13}$C-$CO_2$ time series (Fig. 11, B). As expected, the amplitude of the daily cycle in $\delta^{13}$C-$CO_2$ is negatively correlated with that of $CO_2$ mole fractions, where daily





$\delta^{13}$C-CO$_2$ maxima correspond to day-time minima in CO$_2$ mole fractions. The R$^2$ suggests that 81 % of the variability in $\delta^{13}$C-CO$_2$ could be explained by the variation of CO$_2$ (Fig. 14).

This pattern is generally consistent with CO$_2$ uptake by plants, which preferentially assimilate $^{13}$C depleted CO$_2$, leading to $^{13}$C enriched CO$_2$ in the remaining atmosphere (Ciais et al., 1995a; Bowling et al., 2005; Braden-Behrens et al., 2017). In line with Bowling et al., (2005) and Braden-Behrens et al., (2017), we observe a strong $\delta^{13}$C-CO$_2$ depletion during the build-up of CO$_2$ in the night-time boundary layer, which coincides with Radon build-up, highlighting that terrestrial CO$_2$ fluxes caused this variability at our coastal observation site. This pattern is expected as the ground-level CO$_2$ increase is caused by ecosystem

respiration (around –27.5 ‰ in C3 plant dominated ecosystems) or anthropogenic sources (–26 ‰ to –44 ‰), both of which are strongly depleted in $^{13}$C compared to atmospheric $\delta^{13}$C-CO$_2$ (Vardag et al., 2016; Braden-Behrens et al., 2017).

We observe 19 $\delta^{13}$C-CO$_2$ events (C1-C19). Daily cycles in $\delta^{13}$C-CO$_2$ that are statistically significant but close to the limit of detection are observed at amplitudes of the daily cycle in CO$_2$ mole fractions as low as 2 to 3 ppm (e.g. C6, C7, C15, C17).

This is confirmed by the following back of the envelope calculation. Using the linear regression of the correlation between CO$_2$ mole fractions and $\delta^{13}$C-CO$_2$ (Fig. 14), we can calculate a minimum difference in CO$_2$ mole fractions that would theoretically result in a significant $\delta^{13}$C-CO$_2$ difference of at least twice the $\delta^{13}$C-CO$_2$ measurement uncertainty of ±0.07 ‰ (Fig. 9).

We find that the Delta Ray setup as described and operated at BHD requires a CO$_2$ variability of at least ~2.6 ppm in order to measure a $\delta^{13}$C-CO$_2$ signal that exceeds 0.14 ‰ or twice the analytical uncertainty. Note that this estimation is critically dependent on the isotopic composition of the locally prevailing CO$_2$ source and is therefore specific to the deployment site. The majority of observed daily CO$_2$ cycles exceeds an amplitude of 2.6 ppm (Section 7.2), resulting in a typical amplitude of daily $\delta^{13}$C-CO$_2$ cycles between 0.2 and 0.7 ‰, thereby exceeding the measurement uncertainty by a factor of 3-10, respectively.

Our observations show that the Delta Ray setup is able to resolve most $\delta^{13}$C-CO$_2$ variations at BHD that are associated with terrestrial CO$_2$ fluxes.

This assessment has so far provided critical indicators of instrument performance such as the achievable instrument precision and the limit of the analytical resolution under field deployment conditions. Further to this, we assess the limitations of the

analysis that can be done on the data from the field deployed instrument using Keeling Plot Analysis (KPA). Following the recommendations of Zobitz et al., (2006) for data with small CO$_2$ ranges, we use the Model 1 regression (ordinary least squares) and the standard error as uncertainty of the determined intercept in our KPAs. Bowling et al., (2005) point out that this prevents an erroneous bias of strongly $^{13}$C depleted intercept values at the lowest CO$_2$ ranges, which seems to produce realistic intercept results from our data (e.g. Fig. 15, B). We selected twelve events in the CO$_2$ and $\delta^{13}$C-CO$_2$ time series (Fig. 11, B and C)



ranging from the smallest (2 ppm) to the largest (16 ppm) $CO_2$ mole fraction variations. Selected events include six $CO_2$ peaks and six $CO_2$ troughs, resulting from both night-time boundary layer $CO_2$ build-up and photosynthetic $CO_2$ uptake, respectively.

The $CO_2$ amplitude of these events is well correlated with the amplitude in $\delta^{13}$C-$CO_2$ (Fig. 15, A) with a coefficient of correlation of 0.97. As expected from the findings of Pataki et al. (2003), Bowling et al. (2005) and Zobitz et al., (2006), the

uncertainty of our intercepts increases when the range of $CO_2$ and $\delta^{13}$C-$CO_2$ is small (Fig. 15, C and D). However, it is noteworthy that the data of Pataki et al. (2003) require a minimum $CO_2$ range of 75 ppm to achieve intercept uncertainties of ≤1 ‰, whereas Bowling et al. (2005) apply a lower threshold of ≤40 ppm. In comparison, we find intercept uncertainties of ± 5 ‰ at $CO_2$ ranges of ~2.5 ppm and of ~1 ‰ at $CO_2$ ranges ≥10 ppm (Fig. 15, C), which Zobitz et al., (2006) reported as an acceptable uncertainty level. For further comparison, Pieber et al., (2021) filter their multi-year data to exclude $CO_2$ variations

< 3 ppm and cluster their data for intercept uncertainties of 1, 2, 3 and 4 ‰, which indicates the performance of their instrument is superior to that of the Delta Ray system presented here. Zobitz et al., (2006) model the improvement in the uncertainty of the intercepts with improvement of the measurement precision for isotope ratios. Given the superior measurement precision of the Delta Ray setup of ± 0.07 ‰ (Section 7.2) in comparison to the setup described by Bowling et al., (2005), the uncertainty of the intercepts in our KPAs is smaller as expected. While this proves the gain in the interpretability of measured data with

the instrument performance of the Delta Ray, the comparison with Zobitz et al., (2006) and Pieber et al., (2021) shows that further improvement of instrument precision would be desirable to further improve the usefulness of observations.

The intercepts of the KPAs range between –23 and –33 ‰ with an average of –30 ‰, which is around –3 ‰ more depleted in $^{13}$C than the typical intercept values both Pataki et al., (2003) and Bowling et al., (2005) report. However, our intercept values are in the range that Vardag et al., (2016) report from an urban site and Braden-Behrens et al., (2017) find at a forest site, as

well as in the range that Pieber et al., (2021) report from observations at a background air location.

Figure 16 A and D show KPA intercepts that are in good agreement with the values from Pataki et al. (2003) and Bowling et al., (2005), i.e. the $CO_2$ build-up in the night-time boundary layer on 26 May 2015 and the photosynthetic $CO_2$ draw-down on 2 June 2015 with intercept values of –28 ± 3 ‰ and –28 ± 2 ‰, respectively. The intercepts from both events are in the expected range for C3 plant dominated ecosystems. Indeed, Figure 12 (A and D) show that the air was advected over the

central South Island on 26 May 2015, while it passed over the central North Island on 2 June 2015. The central parts of both islands are predominantly either forest or farmland while urban areas are mostly in the coastal regions, which corroborates the KPA result.

These two events are marked by a $CO_2$ range of 7.5 and 9.2 ppm, respectively (Fig. 12, A and D). In contrast, the photosynthetic uptake event on 29 May 2015 is marked by an intercept of –29 ‰ with a large uncertainty of ± 9 ‰ due to the small $CO_2$ range

of 1.8 ppm (Fig. 12, C). This event directly followed the largest $CO_2$ peak in our time series. Back trajectories for the day-time of 29 May 2015 (not shown) indicate that over the previous 48 h, the measured air was advected from the Tasman sea and has only been in contact with land masses in the last hour before measurement. Despite the large uncertainty of the intercept, its





value of $-29 \pm 9$ ‰ rules out that marine processes have caused the small $CO_2$ decrease, because oceanic $CO_2$ uptake does not discriminate strongly against $^{13}C$.

In contrast, the night-time build-up of $CO_2$ in the early hours of the 29 May 2015 is the largest feature of our time series. For that event, our KPA shows an intercept of $-30 \pm 1$ ‰ (Fig. 16, B), which is slightly more depleted in $^{13}C$ than what we would expect for respiration of C3 plant ecosystems (Pataki et al., 2003; Bowling et al., 2005). Back trajectories from that event show that the air measured during this event has passed over the urban areas of Wellington and Lower Hutt (Fig. 12, B), which likely resulted in the impact of additional urban $CO_2$ emissions in our measurements. Urban $CO_2$ can explain the $^{13}C$ depletion of the

intercept, because besides wood, isotopically depleted natural gas is widely used as fuel for residential heating in the Wellington region.

It is important to keep in mind that the nature of observations by Pataki et al., (2003), Bowling et al., (2005), Zobitz et al., (2006) and Braden-Behrens et al., (2017) is fundamentally different from our study as well as from that of Vardag et al., (2016) and Pieber et al., (2021). Due to the remote location of our study site and that of Pieber et al., (2021), observed $CO_2$ and $\delta^{13}C$-

$CO_2$ variations result from a spatio-temporal integration of multiple $CO_2$ processes and different ecosystem types along the air flow path. In comparison, the observations of Pataki et al., (2003), Bowling et al., (2005) and Braden-Behrens et al., (2017) were made within one ecosystem, while that of Vardag et al., (2016) were made in an urban environment. Resulting differences between the intercepts derived in these studies are therefore expected.

**7.4 $\delta^{18}O$-$CO_2$ observations of the Delta Ray during deployment at BHD**

Over the course of our study period, the $\delta^{18}O$-$CO_2$ measurements vary around a baseline value of $+1.1$ ‰ (Fig. 11). This value is in the expected range for a coastal site in the mid latitudes of the Southern Hemisphere, and it is in good agreement with observations from the Commonwealth Scientific and Industrial Research Organisation (CSIRO) at the Cape Grim Observatory (CGO) (Francey and Tans, 1987; Welp et al., 2011). In comparison to $\delta^{13}C$-$CO_2$, the $\delta^{18}O$-$CO_2$ time series does not show a

strong correlation with $CO_2$ mole fractions (Fig. 14). The $\delta^{18}O$-$CO_2$ time series shows 22 distinct events with amplitudes that range between 0.1 ‰ and 1.5 ‰, labelled O1 – O22 in Figure 11. All events except O8 occur during northerly wind conditions and are most pronounced during low wind speed. All events except O5 and O20 occur during day-time, while $\delta^{18}O$-$CO_2$ typically declines as Radon levels increase with the build-up of the night-time boundary layer (e.g. O2, O4, O6), when the observed daily $\delta^{18}O$-$CO_2$ cycle is a signal of the terrestrial biosphere. Atmospheric $CO_2$ undergoes oxygen isotope exchange

with $^{18}O$-enriched leaf water (Francey and Tans, 1987; Farquhar et al., 1993; Welp et al., 2011; Cernusak et al., 2016), which is modulated by stomatal conductance that is generally high at day-time and low at night-time (Caird et al., 2007). Most peaks in the $\delta^{18}O$-$CO_2$ record are therefore a result of photosynthetic activity of the terrestrial biosphere. This explains the lack of a $\delta^{18}O$-$CO_2$ signal during S1, S2 and S3, when the air was advected over the ocean (Fig. 13) and had limited (S1, S3) or no (S2) contact with the terrestrial biosphere in the recent past.





Changes in the $\delta^{18}O$-$CO_2$ record suggest that the measured $CO_2$ has been in isotopic exchange with different water bodies of different isotopic compositions. Baisden et al., (2016) show the spatial variability of the isotopic composition of precipitation in New Zealand. In general, $\delta^{18}O$ in precipitation becomes more depleted with i) increasing distance to the equator (latitudinal gradient), ii) increasing distance to the precipitation source (Ocean water) and iii) increasing altitude (Dansgaard, 1964; Baisden et al., 2016). In a very simplified approach, we can assume that the spatial isotope pattern of the precipitation creates

a corresponding pattern in the isotopic composition of leaf and soil water bodies, which will impact on $\delta^{18}O$-$CO_2$ during isotope exchange accordingly (Farquhar et al., 1993; Cernusak et al., 2016).

An example of this can be observed during 2 to 3 June 2015 and 6 June 2015, when $\delta^{18}O$-$CO_2$ appears 0.3 ‰ to 0.6 ‰ more depleted than average, indicating that this $CO_2$ has been in isotopic exchange with $^{18}O$ depleted leaf and soil water. In fact, HYSPLIT back-trajectories show that air masses measured during this period were predominantly advected from inland

regions and higher altitudes (Fig. 12, D), where the $\delta^{18}O$ of the precipitation is more depleted than in other regions (Baisden et al., 2016).

Interestingly, the night-time events O5 and O20 show an $^{18}O$ enrichment of 0.7 to 1.0 ‰, which is accompanied by a simultaneous $^{13}C$ depletion of ~0.6 ‰ and increased $CO_2$ mole fractions of 7 to 10 ppm. This combined pattern furthermore coincides with peaks in Radon (Fig. 11). Back trajectories from O20 (not shown) reveal that the measured air has been advected

from the West Coast of the South Island, from where it passed over alpine areas as well as over the city of Wellington before it was measured at BHD. Schumacher et al., (2011) report $^{18}O$ enrichment and $^{13}C$ depletion in $CO_2$ derived from wood combustion. It is thus likely that the increased $CO_2$ originated from a combination of ecosystem respiration and anthropogenic combustion processes. In contrast, we observe the opposite $\delta^{18}O$-$CO_2$ trend during the largest $CO_2$ peak in our time series (29 May 2015), which might suggest that the relative contribution of each $CO_2$ source category is different for O5 and O20 in

comparison to the major $CO_2$ event on the 29 May 2015.

Our observations show that the Delta Ray is capable of resolving small changes in $\delta^{18}O$-$CO_2$ and that these measurements enable further analysis of anthropogenic and ecosystem processes along the pathways of advected air masses.

**8. Capabilities and limitations of Delta Ray to resolve small variations of Southern Ocean air events at BHD**

**8.1 Variability of $CO_2$ mole fractions during southerlies**

Figure 17 zooms into the measurement data from the three southerly periods S1, S2 and S3. Delta Ray and Siemens data show good agreement throughout the events and differences of the event averages (calculated as Delta Ray – Siemens) of –0.03 ppm, –0.08 ppm and +0.04 ppm $CO_2$, for S1, S2 and S3, respectively, which is within the compatibility goal of the WMO for S1 and S3, but not for S2. The Delta Ray $CO_2$ data for S2 include one strongly elevated $CO_2$ value towards the end of the

event. If this data pair was removed, the disagreement between Delta Ray and Siemens during S2 would increase to –0.11





ppm. Both analysers show S2 with the lowest $CO_2$ mole fractions, which seems plausible given the potential for added terrestrial $CO_2$ during both S1 and S3 (Fig. 13). A possible explanation for the variability between measurements is the different measurement schedules both analysers operate on. While the Delta Ray observations follow a measurement schedule containing only 22 % air measurements per cycle (Table 2), the measurement schedule of the Siemens during southerlies

measures air during 50 % of the time and makes thus more than twice as many observations, but not necessarily at the same time as the Delta Ray. The difference in timing may explain some differences between the measurements of both systems. However, we would expect the differences to be minimal during southerly events and especially during steady intervals, as is the case for S2.

## 8.2 Accuracy of isotope measurements of Delta Ray, assessed using Southern Ocean events at BHD


Unlike the case for $CO_2$ mole fractions, we have no means to assess the accuracy of the isotope data from the Delta Ray with independent observations, because the flask sampler at BHD was not operational during the Delta Ray deployment. As a next best solution, we compare the Delta Ray isotope data with IRMS-based data that are available from BHD ($\delta^{13}C$-$CO_2$ only) as well as from the Cape Grim Observatory (CGO) in Tasmania, Australia ($\delta^{13}C$-$CO_2$ and $\delta^{18}O$-$CO_2$) from times adjacent to the

Delta Ray time series campaign. We think that this comparison is feasible for two reasons: i) the isotope values for QC-1 were assigned using the same GC-IRMS setup and scale realisation that is also used to make the $\delta^{13}C$-$CO_2$ measurements in the flask samples from BHD (Section 2.2). Therefore, we think that potential calibration offsets between the Delta Ray, NIWA's GC-IRMS and CSIRO's observations at CGO should be minimal. ii) during similar phases of the seasonal cycle, the observable difference between stations in the Southern Hemisphere at comparable latitudes is very small. For example, the seasonal $\delta^{13}C$-

$CO_2$ cycle at CGO is in the order of 0.05 ‰ (Allison and Francey, 2007). The very small $\delta^{13}C$-$CO_2$ variability in Southern Ocean air justifies the comparison of baseline observations from multiple sites (e.g. Ciais et al., (1995a) for $\delta^{13}C$-$CO_2$, Welp et al., (2011) for $\delta^{18}O$-$CO_2$, Stephens et al., (2013) for $CO_2$).

GC-IRMS measurements are available from flask samples taken at BHD during two southerlies on 28 April 2015 and 23 June 2015, where the latter fulfilled steady interval criteria. Furthermore, $\delta^{13}C$-$CO_2$ and $\delta^{18}O$-$CO_2$ observations were made at CGO

on 13 and 29 May 2015 as well as on 9 and 24 June 2015. We use a linear interpolation between the discrete observations from both BHD and CGO to estimate $\delta^{13}C$-$CO_2$ and $\delta^{18}O$-$CO_2$ values for the periods of S1, S2 and S3. We find good agreement in the interpolated $\delta^{13}C$-$CO_2$ values between samples from BHD and CGO of <0.02 ‰ (green lines in Figure 17), which corroborates our approach to compare measurements made with the Delta Ray to observations made in glass flasks on different times and at a different station in the Southern Ocean. However, the comparison for S1 and S3 is compromised as these events

did not fulfil baseline criteria.

We determine the agreement between the Delta Ray and interpolated IRMS-based measurements (calculated as Delta Ray – IRMS) during S2 as –0.10 ‰ for $\delta^{13}C$-$CO_2$ and –0.20 ‰ for $\delta^{18}O$-$CO_2$ (Fig. 17). For $\delta^{13}C$-$CO_2$, this difference accounts for





twice the amplitude of the seasonal cycle seen at CGO of ~0.05 ‰ (Allison and Francey, 2007), while the range is smaller than the amplitude of the seasonal $\delta^{18}O\text{-}CO_2$ cycle observed at CGO during 2015 of ~0.3 ‰. In the light of the compatibility

goals of 0.01 ‰ for $\delta^{13}C\text{-}CO_2$ and 0.05 ‰ for $\delta^{18}O\text{-}CO_2$ (WMO-GAW, 2019), this requires further investigation. In the case of $\delta^{13}C\text{-}CO_2$, this difference accounts for twice the measurement uncertainty, while it exceeds the measurement uncertainty of $\delta^{18}O\text{-}CO_2$ by a factor of 4. We can think of a few possible reasons for this difference.

Additional $\delta^{13}C\text{-}CO_2$ observations from BHD made in flask samples from 22 July 2015 and 3 September 2015 show $\delta^{13}C\text{-}CO_2$

values of $-8.54 \pm 0.03$ ‰ and $-8.56 \pm 0.02$ ‰, respectively. While these $\delta^{13}C\text{-}CO_2$ values are in the same range as the observations made with the Delta Ray system, they are significantly more depleted in $^{13}C$ than all other flask sample observations from BHD and CGO during 2015. It seems thus possible that the $\delta^{13}C\text{-}CO_2$ values observed during S1, S2 and S3 are a true atmospheric $\delta^{13}C\text{-}CO_2$ signal that is different from the observations at CGO.

Furthermore, this difference may partly be explained by the uncertainty of the target value assignment to the working standard

QC-1, which accounts for 0.07 ‰ for $\delta^{13}C\text{-}CO_2$ and 0.04 ‰ for $\delta^{18}O\text{-}CO_2$ (Table 1). An offset in the determination of the target value of QC-1 would result in a corresponding shift of the entire Delta Ray data set. While the uncertainty in the value assignment of QC-1 may explain up to 70 % of the difference to the interpolated flask values for $\delta^{13}C\text{-}CO_2$, it can only explain 20 % of that difference in $\delta^{18}O\text{-}CO_2$. Further tests would be required to scrutinise the scale realisation effect but are out of scope for this study.

Another aspect that potentially creates an offset in our Delta Ray setup is the systematic difference between the measurements made in air samples at ambient pressure and the measurement of reference gases that are supplied by high-pressure cylinders and delivered to the system at above-ambient pressures. Therefore, measurements of reference and sample gases are made at systematically different pressure regimes. We did not evaluate the impact of different gas supply pressures on the resulting isotope data. However, $\delta^{13}C\text{-}CO_2$ measurements using air from glass flasks showed that $\delta^{13}C\text{-}CO_2$ was drifting with lowering

pressure in the flask. Even though the Delta Ray should tolerate inlet pressures between 700 and 1200 mbar (Thermo, 2014), our experiments in that pressure window showed isotope effects with a magnitude that could explain the observed difference. Because we set all pressure regulators of the QC gases to identical pressures, a measurement artefact due to pressure differences would affect the measurements of all QC gas cylinder with comparable magnitude. Indeed, our QC gas measurements show similar offsets compared to their assigned target values (Table 3). It is thus possible that an unquantified pressure bias caused

differences between measurements of ambient air and QC gases from cylinders. Further tests with direct comparisons between Delta Ray and IRMS-based methods that explore the effect of different gas delivery pressures are needed to assess factors limiting the accuracy of the Delta Ray.



### 8.3 Limitations of $\delta^{13}$C-CO$_2$ and $\delta^{18}$O-CO$_2$ observations of the Delta Ray during Southern Ocean events at BHD

The WMO has formulated challenging compatibility goals for the analytical performance of instruments to measure CO$_2$, $\delta^{13}$C-CO$_2$ and $\delta^{18}$O-CO$_2$ in Southern Hemispheric baseline air (WMO-GAW, 2019). However, the specifications of the Delta Ray instrument for isotope ratio measurements exceed the compatibility goals by a factor of 2 to 5. The very small atmospheric variation observable during southerlies at BHD represent a challenging environment to assess the capability and limitations of the instrument.

Isotope observations during S1, S2 and S3 highlight the limitations of the Delta Ray to resolve small atmospheric variations. We found the average $\delta^{13}$C-CO$_2$ values of S1 with $-8.51 \pm 0.02$ ‰ to be by 0.05 ‰ more depleted in $^{13}$C than the values during the steady interval S2 with $\delta^{13}$C-CO$_2$ of $-8.46 \pm 0.05$ ‰. Likewise, S3 with a $\delta^{13}$C-CO$_2$ of $-8.48 \pm 0.04$ ‰ was by 0.02 more depleted in $^{13}$C than S2. However, considering the analytical uncertainties of these observations, we are unable to resolve the differences between the events at significant levels. While it seems plausible that S1 and S3 have more negative $\delta^{13}$C-CO$_2$ 
values due to the potential for additional terrestrial CO$_2$, analysing differences of this magnitude does not provide robust results. The same limitations apply to the $\delta^{18}$O-CO$_2$ results from S1, S2 and S3. An improvement of the achievable measurement precision would be required to resolve the variability or to assess the similarity of isotope ratios during Southern Ocean baseline events.

### 9. Summary and conclusion

We tested the Delta Ray analyser in the laboratory and at BHD, our observatory for Southern Ocean baseline air. We developed a calibration scheme for the Delta Ray system that is different from that recommended by the manufacturer. Our calibration scheme includes measurements of two quality control gases in every measurement sequence for instrument calibration and assessment, fulfilling the Principle of Identical Treatment (PIT). We achieved a long-term reproducibility of 0.07 ‰ for $\delta^{13}$C-CO$_2$, 0.06 ‰ for $\delta^{18}$O-CO$_2$ and 0.03 ppm for CO$_2$ mole fractions. We demonstrated that our changes to the calibration approach 
sufficiently controlled instrument linearity, which was reported as problematic in previous studies (Braden-Behrens et al., 2017; Flores et al., 2017). However, our calibration technique limited the length of time the system was able to measure air within each sequence. A preferred method of operation would be to enable the use of CO$_2$ in air standards instead of Ref-1 and Ref-2 at the operator's discretion. Especially when deployed at sites of low CO$_2$ mole fraction variability such as BHD, the inbuilt capability for dynamic mixing of Ref-1 and Ref-2 seems unnecessary. The reliance on Ref-1 and Ref-2 adds logistical 
complications due to the strict quality requirements on CO$_2$-free air. We demonstrated the sensitivity of the system to different commercial CO$_2$-free air suppliers and find that commercial or home-made purified air delivered the most accurate results. The deployment period at BHD included a range of atmospheric advection patterns, resulting in daily CO$_2$ cycles of variable amplitude, periods with variable degree of terrestrial influence on CO$_2$ and reoccurring Southern Ocean events with very little variability of CO$_2$ and its isotope ratios (S1, S2 and S3). We think that the deployment at BHD with its very small variation in



CO$_2$ represented a challenging environment to assess the instrument performance of the Delta Ray under field deployed conditions.

Overall, we find the CO$_2$ mole fraction measurements made with the Delta Ray in good agreement with our well-established system at BHD (Brailsford et al., 2012; Stephens et al., 2013), over mole fraction changes between 2 to 16 ppm (Section 7.1, Fig. 11). We find 39 % and 21 % of the data pairs in agreement with the WMO compatibility goals of 0.1 ppm and 0.05 ppm for the Northern and Southern Hemisphere, respectively. Existing differences in CO$_2$ mole fraction measurements likely originate from different data reduction and averaging intervals from both instruments. We expect that synchronising the timing would have further improved the instrument agreement. Our Delta Ray setup relied on a one-point calibration for CO$_2$ mole fractions only, while a two-point calibration is recommended by the manufacturer for more accurate measurements.

While the instrument performance did not meet the WMO network compatibility goals of 0.01 ‰ for $\delta^{13}$C-CO$_2$ and 0.05 ‰ for $\delta^{18}$O-CO$_2$, it did meet the WMO expanded compatibility goal of 0.1 ‰ for both $\delta^{13}$C-CO$_2$ and $\delta^{18}$O-CO$_2$. In line with previous studies, we found the uncertainty of data analysis to be inversely scaled to the amplitude of CO$_2$ changes (Pataki et al., 2003; Bowling et al., 2005). In comparison to these studies, the superior instrument precision of the Delta Ray enables the analysis of smaller CO$_2$ signals with smaller amplitude. We demonstrated the capability of Keeling Plot Analysis (KPA) on selected events in the Delta Ray time series to provide intercepts with uncertainty of ~1 ‰ when CO$_2$ signals exceed 10 ppm. KPA on smaller CO$_2$ signals was possible if larger intercept uncertainty was tolerable. However, we found the limit of resolution at ~3 ppm, where the Delta Ray was capable to resolve variations in isotope ratios that were in line with expected $\delta^{13}$C-CO$_2$ and $\delta^{18}$O-CO$_2$ signals based on trajectories of air advection and associated biogeochemical processes. For robust analysis of our data, however, further improvement of the measurement precision to ≤0.01 ‰ would be desirable to meet the WMO network compatibility goal and to distinguish the variability in Southern Ocean baseline air.

## 10. Data availability

Date are available upon request.

## 11. Author contributions

PS, GWB, RCM and IS conceptualised the study and developed the method, MM supplied the instrument, PS, SMF, BB and JMG analysed the data, SN and PK curated and provided additional data, PS wrote the manuscript with contributions from all authors.



## 12. Competing Interests

The authors declare no competing interests.

## 13. Acknowledgements

This work was funded by the National Institute of Water and Atmospheric Research through the Greenhouse Gases, Emissions and Carbon Cycle Science Programme. Further support was received through the CarbonWatch NZ research programme
(C01X1817) and Marsden funded project NIW-1704. We gratefully acknowledge the support from Thermo in supplying us with a Delta Ray demo instrument. The Australian Bureau of Meteorology and CSIRO are thanked for their long-term support of the Cape Grim station and CSIRO GASLAB.

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



**Table 1: Two pure CO₂ gases used as reference gases (Marsden, Kapuni) were calibrated to VPDB realisation of Lowe et al. (1994). QC-1, QC-2 and QC-3 served as air standard and as target gases, respectively, while QC-4 and QC-5 were used during laboratory tests. Mole fractions and isotope ratios are provided with the 1σ uncertainty of their measurement.**

| name | function | cylinder | gas matrix | CO₂ [ppm] | u_CO₂ [ppm] | $\delta^{13}C$ [‰] | u_$\delta^{13}C$ [‰] | $\delta^{18}O$ [‰] | u_$\delta^{18}O$ [‰] | gas provider |
|---|---|---|---|---|---|---|---|---|---|---|
| Kapuni | Ref-1 | 46 | pure CO₂ | 100 % | | −13.748 | 0.1 | −11.695 | 0.1 | BOC |
| Marsden | Ref-2 | 26 | pure CO₂ | 100 % | | −32.768 | 0.1 | −32.518 | 0.1 | Air Liquide |
| CO₂-free air | carrier, BHD deployment | CB11048 | purified natural air | <1 | | | | | | Scott Marrin |
| CO₂-free air | carrier, experiment | | purified natural air | <1 | | | | | | NIWA |
| CO₂-free air | carrier, experiment | | synthetic N₂/O₂ mixture, no Ar | <1 | | | | | | BOC |
| QC-1 | standard | CB 09725 | natural air | 400.43 | ±0.09 | −8.54 | ±0.07 | −0.62 | ±0.04 | NIWA |
| QC-2 | target 1 | 43410 | natural air with CO₂ spike | 497.71 | ±0.07 | −13.42 | ±0.02 | −6.11 | ±0.07 | NIWA |
| QC-3 | target 2 | CB10800 | natural air | 396.03 | ±0.06 | −8.32 | ±0.04 | +0.31 | ±0.01 | NIWA |
| QC-4 | test gas | 6235 | natural air | 402.78 | ±0.04 | −8.47 | ±0.01 | +0.17 | ±0.02 | NIWA |
| QC-5 | test gas | 6260 | natural air with CO₂ spike | 424.9 | ±1.5 | −9.82 | ±0.01 | −1.35 | ±0.04 | NIWA |



**Table 2: Timing used for blocks in measurement sequence, resulting gas consumptions, resulting requirements for cylinder replacements per year and relative time each gas was measured.**

| gas | time [s] | measure/flush | consumption [L/d] | replacements / a | relative use time [%] | relative measurement time [%] |
|---|---|---|---|---|---|---|
| Ref-1 + carrier | 150 | flush | 19 | 2 | 16.7 | 11.1 |
| Ref-1 + carrier | 300 | measure | | | | |
| QC-1 | 150 | flush | 32 | 3 | 27.8 | 22.2 |
| QC-1 | 200 | measure | | | | |
| QC-1 | 200 | measure | | | | |
| QC-1 | 200 | measure | | | | |
| air sample | 150 | flush | 32 | – | 27.8 | 22.2 |
| air sample | 200 | measure | | | | |
| air sample | 200 | measure | | | | |
| air sample | 200 | measure | | | | |
| QC-2, QC-3 | 150 | flush | 32 | 3 | 27.8 | 22.2 |
| QC-2, QC-3 | 200 | measure | | | | |
| QC-2, QC-3 | 200 | measure | | | | |
| QC-2, QC-3 | 200 | measure | | | | |
| Ref-1 + carrier | 150 | flush | – | next cycles 1st measurement | – | |
| Ref-1 + carrier | 300 | measure | | | | |








**Table 3: Target values (GC and GC-IRMS) and Delta Ray measurements for QC-1, QC-2 and QC-3 as determined during operation at BHD and differences as Delta Ray minus GC-IRMS and values for QC-2 after correction for the offset in QC-1 minus the GC-IRMS target value.**

| | QC-1 | QC-2 | QC-3 |
|---|---|---|---|
| system | $\delta^{13}C\text{-}CO_2$ [‰] | $\delta^{13}C\text{-}CO_2$ [‰] | $\delta^{13}C\text{-}CO_2$ [‰] |
| GC-IRMS | $-8.54 \pm 0.07$ | $-13.42 \pm 0.02$ | $-8.32 \pm 0.04$ |
| Delta Ray | $-8.74 \pm 0.06$ | $-13.60 \pm 0.12$ | $-8.54 \pm 0.07$ |
| Delta Ray – GC-IRMS | $-0.20 \pm 0.13$ | $-0.18 \pm 0.14$ | $-0.22 \pm 0.11$ |
| QC-1-corr – GC-IRMS | | $+0.02$ | $-0.02$ |

| system | $\delta^{18}O\text{-}CO_2$ [‰] | $\delta^{18}O\text{-}CO_2$ [‰] | $\delta^{18}O\text{-}CO_2$ [‰] |
|---|---|---|---|
| GC-IRMS | $-0.62 \pm 0.04$ | $-6.11 \pm 0.07$ | $+0.31 \pm 0.01$ |
| Delta Ray | $-1.22 \pm 0.07$ | $-6.80 \pm 0.14$ | $-0.31 \pm 0.07$ |
| Delta Ray – GC-IRMS | $-0.60 \pm 0.11$ | $-0.69 \pm 0.21$ | $-0.62 \pm 0.08$ |
| QC-1-corr – GC-IRMS | | $-0.08$ | $-0.04$ |

| system | $CO_2$ [ppm] | $CO_2$ [ppm] | $CO_2$ [ppm] |
|---|---|---|---|
| GC | $400.43 \pm 0.09$ | $497.71 \pm 0.07$ | $396.03 \pm 0.06$ |
| Delta Ray | $400.88 \pm 0.19$ | $498.41 \pm 0.22$ | $396.54 \pm 0.12$ |
| Delta Ray – GC | $0.45 \pm 0.28$ | $0.71 \pm 0.29$ | $0.51 \pm 0.18$ |
| QC-1-corr – GC | | $0.27$ | $0.01$ |






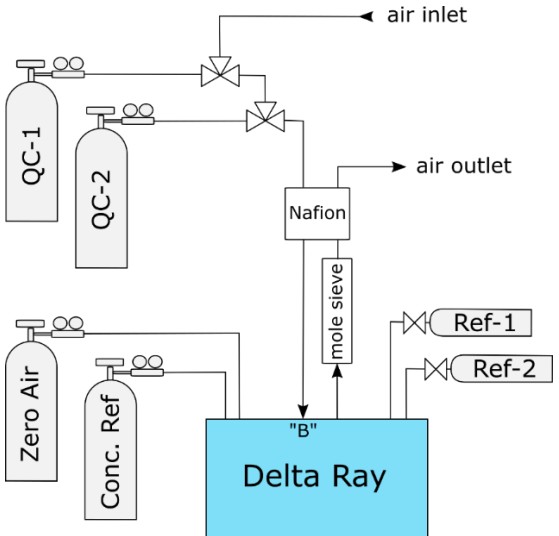

**Figure 1: Delta Ray setup as deployed at BHD.**






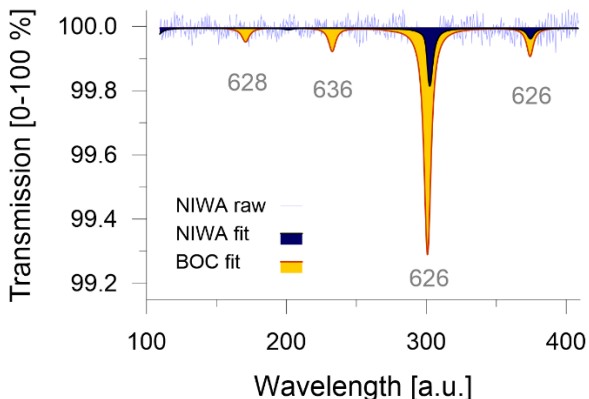

**Figure 2: Size of CO₂ blank in CO₂-free air, transmission [%] over "artificial units [a.u.]", provided by Delta Ray. Raw (light blue) and fitted (dark blue) data from CO₂-free air made at NIWA in comparison with fitted spectra of BOC's Ultra-zero grade air (orange).**







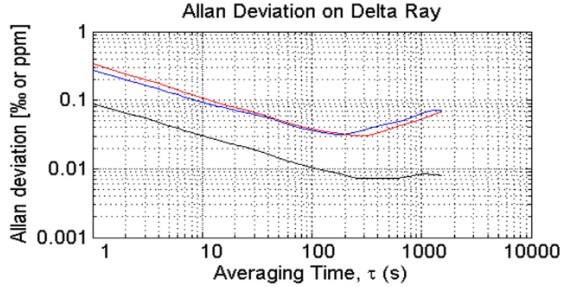

**Figure 3: Allan deviation determined in QC-1: blue = $\delta^{13}$C, red = $\delta^{18}$O, black = CO₂, determined during the 2015 campaign.**






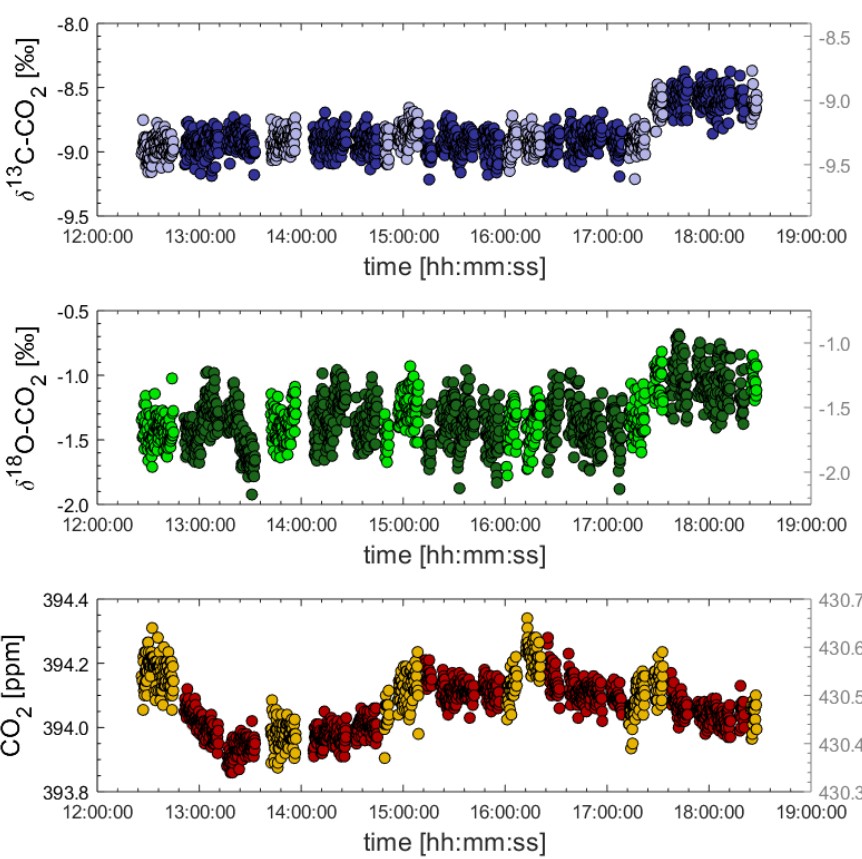

**Figure 4: Sudden and synchronous shift in both $\delta^{13}$C-CO$_2$ (top) and $\delta^{18}$O-CO$_2$ (middle) affecting the measurement of two QC gases during laboratory tests. Shifts in isotope traces are not reflected in CO$_2$ mole fractions (bottom). Causes of the isotope shifts are unknown.**







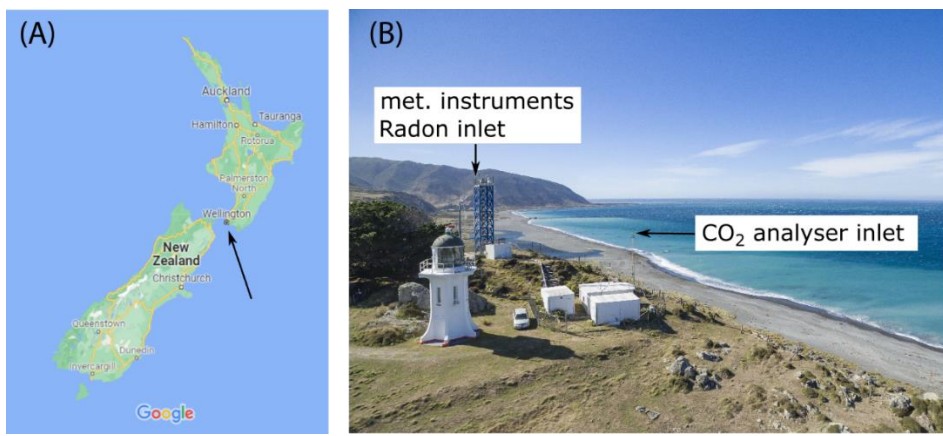

**Figure 5: Left: Arrow highlights the location of BHD in New Zealand (© Google Maps 2021). Right: Aerial photograph of BHD (provided by Dave Allen, NIWA).**









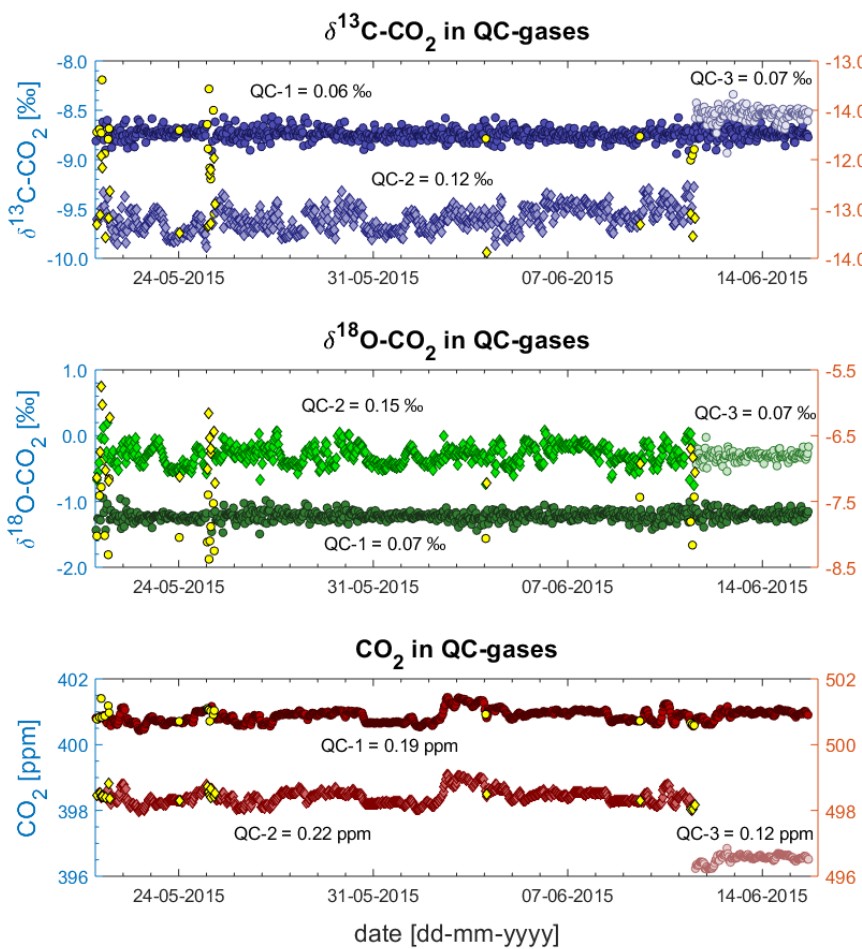

**Figure 6: QC gas data as measured against Ref-1 during deployment at BHD, $\delta^{13}$C-CO$_2$ (top), $\delta^{18}$O-CO$_2$ (middle) and CO$_2$ mole fractions (bottom). QC-1 and QC-3 on left y-axes, QC-2 on right y-axes.**





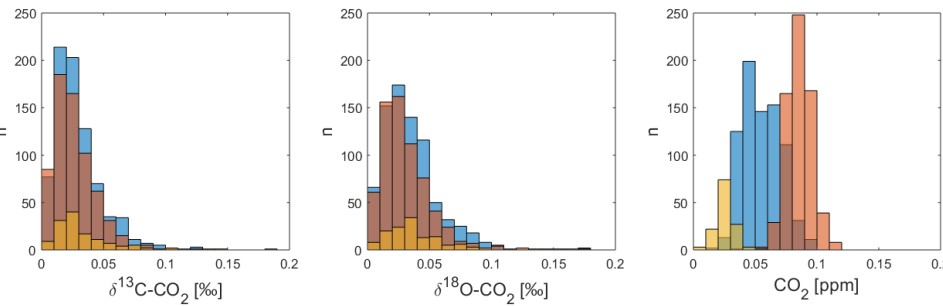

**Figure 7: Internal reproducibility as the standard deviation (1σ) of the QC gas block averages in each sequence for $\delta^{13}$C-CO₂ (left), $\delta^{18}$O-CO₂ (middle) and CO₂ mole fractions (right), outliers excluded. QC-1 in blue, QC-2 in red and QC-3 in yellow.**






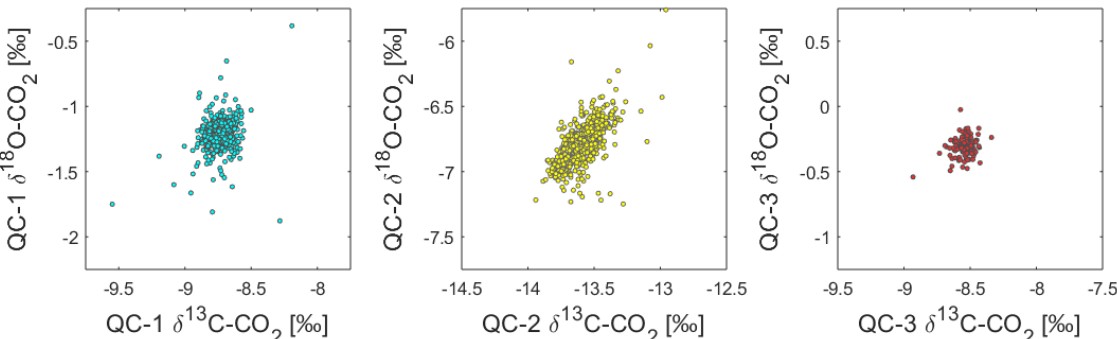

**Figure 8: Correlation of $\delta^{13}$C-CO₂ and $\delta^{18}$O-CO₂ values in QC-1 (left), QC-2 (middle) and QC-3 (right) before outlier removal and correction for QC-1.**










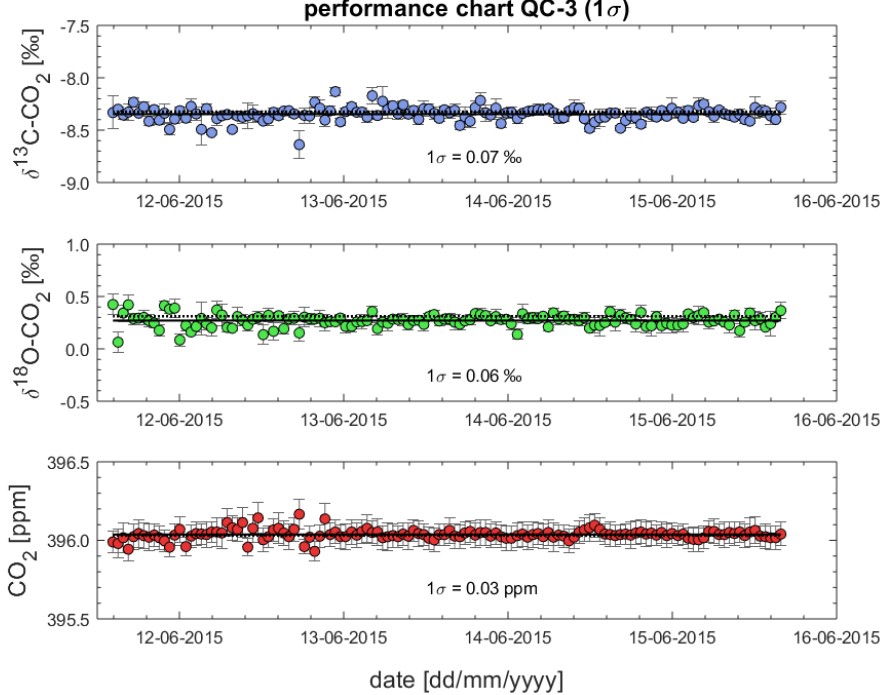

Figure 9: Sequence averages for processed $\delta^{13}$C-CO$_2$ (top), $\delta^{18}$O-CO$_2$ (middle) and CO$_2$ (bottom) in QC-3. The standard deviation (1σ) is a measure of the achievable analytical precision.








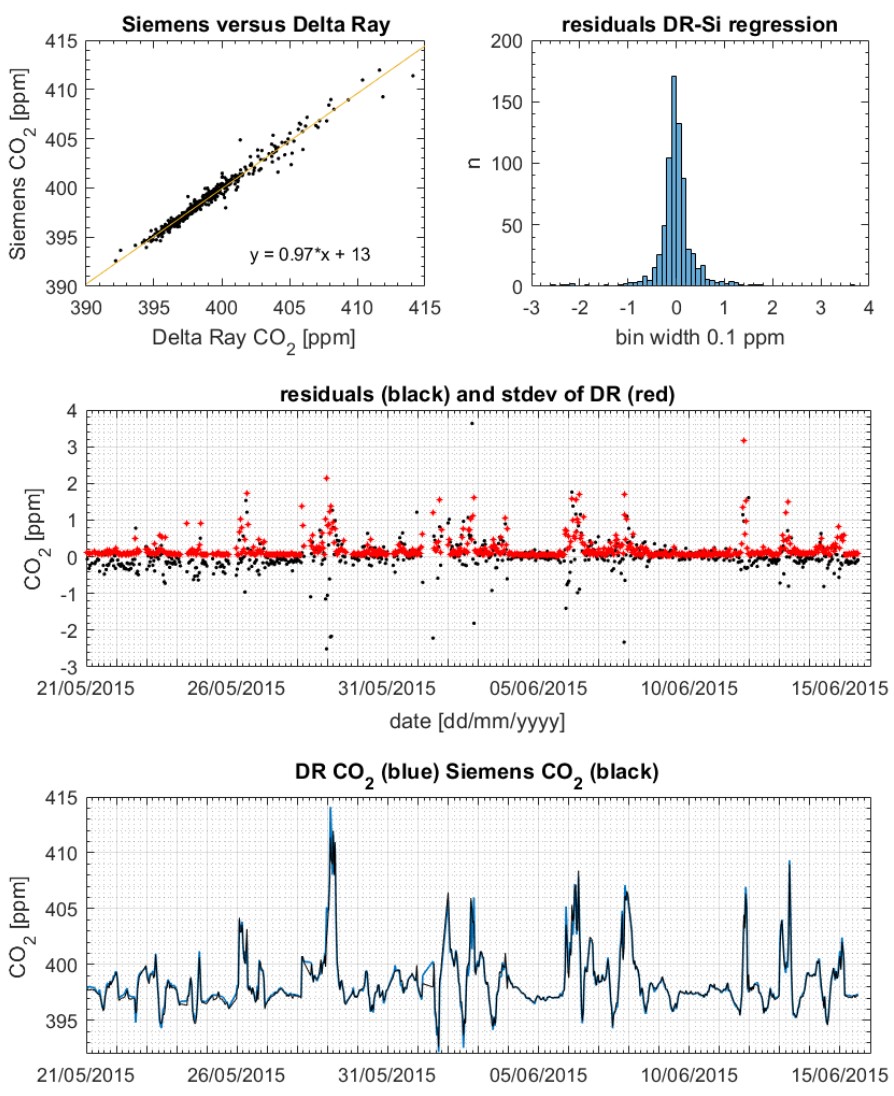

**Figure 10: Top left: slope and scatter of the regression Siemens (Si) versus Delta Ray (DR). Top right: histogram of residuals (top right). Middle: residuals (black) and 1σ standard deviation of the DR measurements (red). Bottom: CO₂ mole fractions from DR (blue) and Siemens (black).**



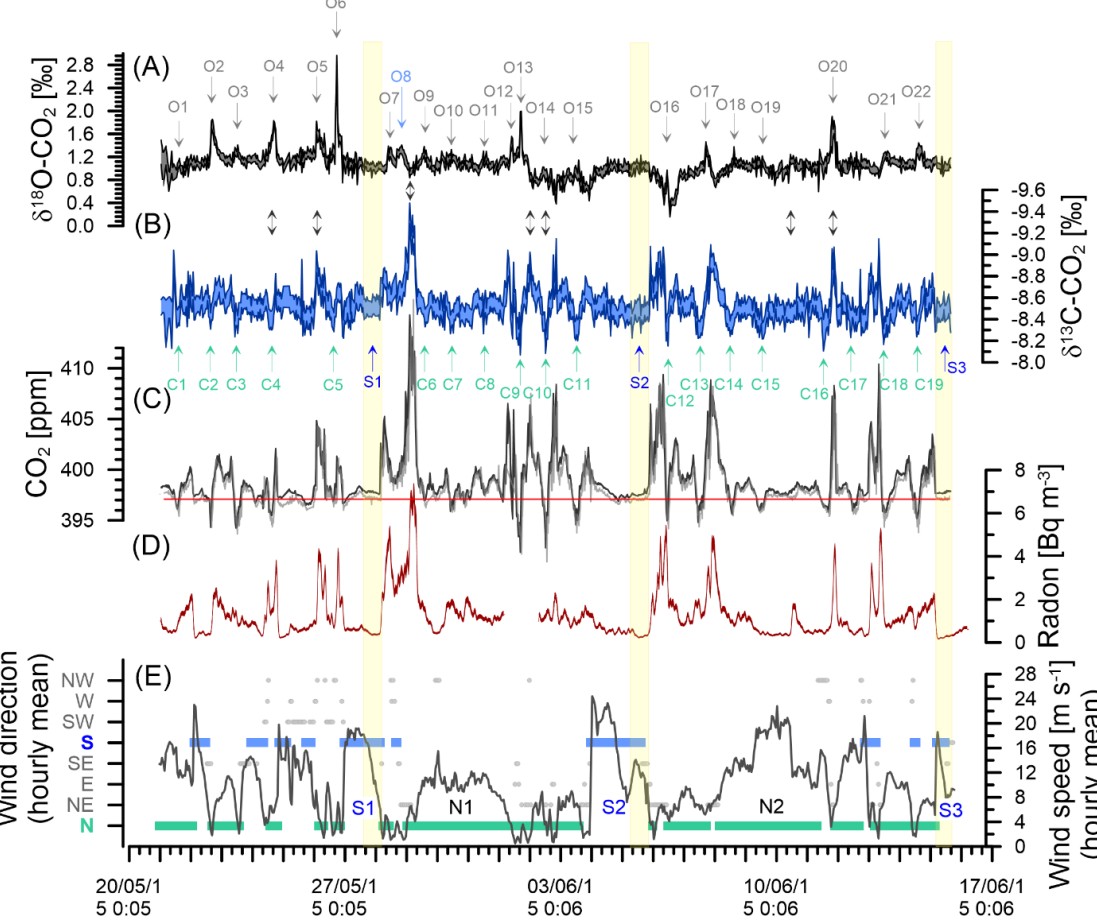

**Figure 11: Observations during the deployment at BHD. Delta Ray time series of $\delta^{18}O$-$CO_2$ (A) and $\delta^{13}C$-$CO_2$ with inverted y-axis (B). $CO_2$ mole fractions measured with Delta Ray (black) and Siemens (grey) with average of the Siemens analyser for baseline $CO_2$ measured during S2 indicated in red (C). Coloured shading in (A), (B) and (C) indicates the standard deviation of the measurement averages of the Delta Ray measurements. Events in the $\delta^{18}O$-$CO_2$ and $\delta^{13}C$-$CO_2$ time series are numbered from O1 to O22 and C1 to C19, respectively. (D) shows the Radon time series and (E) hourly averages of meteorological data. Southerly and northerly wind events are highlighted in blue and green, respectively. Wind speed is displayed by the black line. The yellow boxes highlight the most stable periods of the three significant southerly events S1, S2 and S3.**



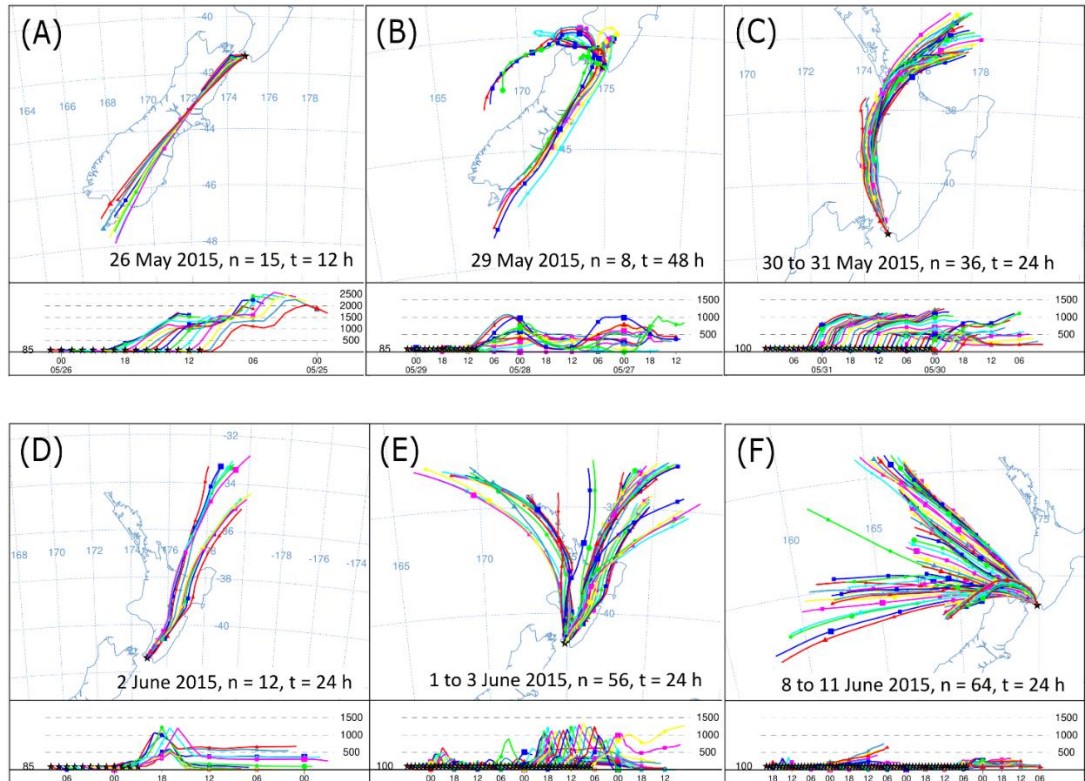

**Figure 12: HYSPLIT back trajectories from six events in the Delta Ray time series. Each panel shows the history of air that was measured at the stated date. Intervals between trajectories are 1 h, n indicates the number of trajectories in the panel, t represents the trajectory length in hours.**





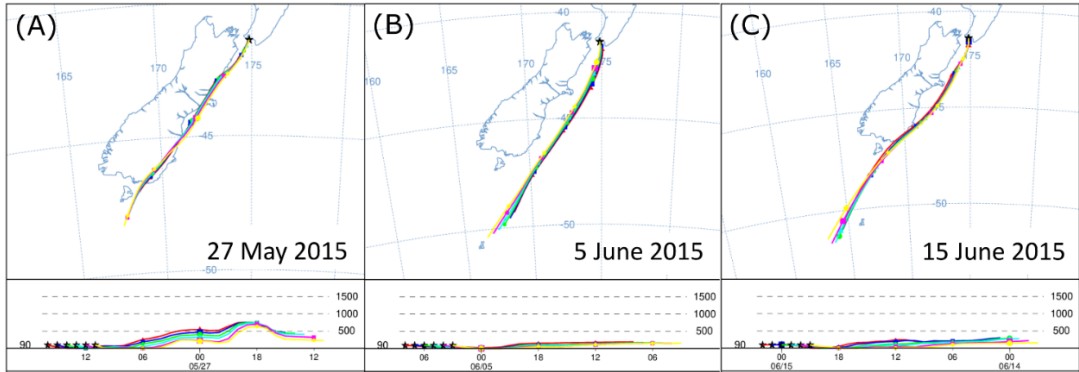

**Figure 13: HYSPLIT back trajectories (24 h) for the air measured during S1, S2 and S3, with a 1 h interval between displayed**
**trajectories. The selected timing is consistent with the final 6 h of each event displayed in Figure 17 with the lowest Radon counts.**









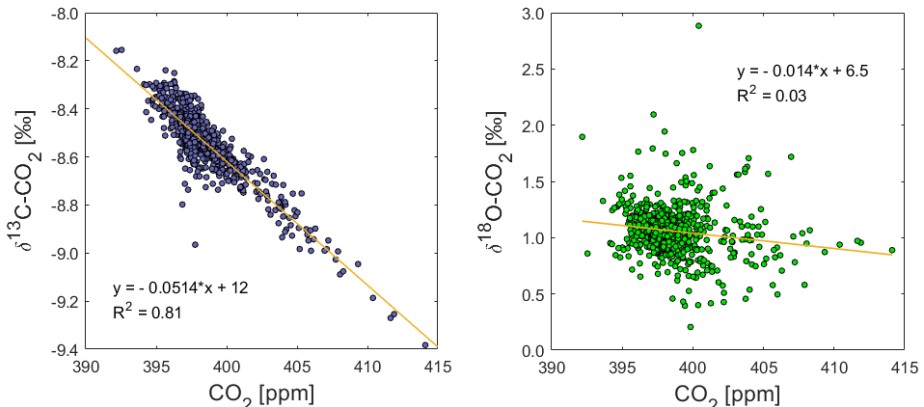

**Figure 14: Correlation between $CO_2$ and $\delta^{13}C\text{-}CO_2$ (left) and $\delta^{18}O\text{-}CO_2$ (right) in air (n = 791).**







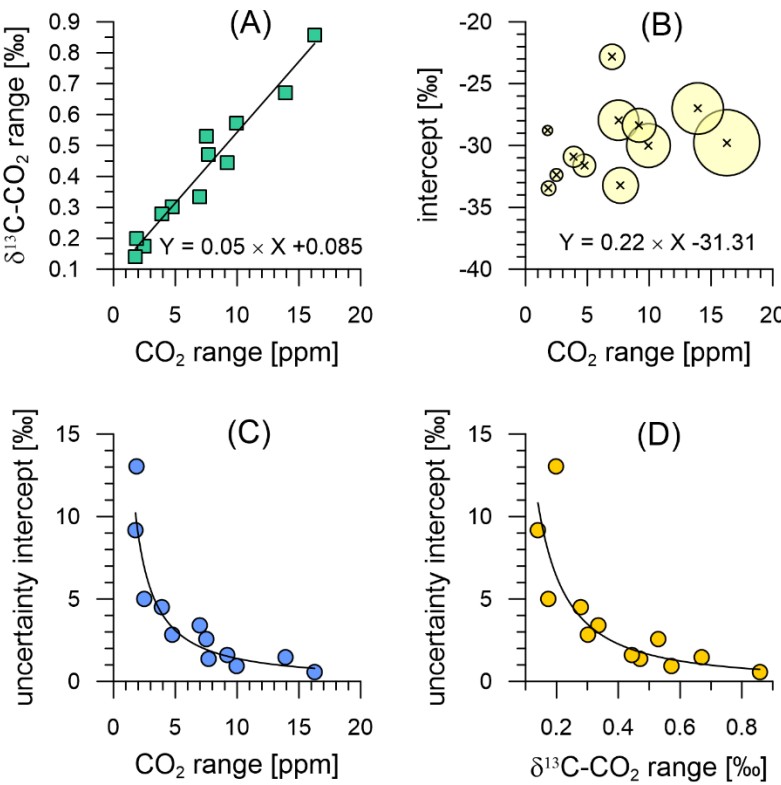

**Figure 15: Relationship between the variations in $CO_2$ and $\delta^{13}C$-$CO_2$ of the twelve events used for KPA (A), as well as their intercept as function of the covered $CO_2$ range (B). The bubble size in (B) scales with the range in $\delta^{13}C$-$CO_2$ covered within the KPA event. Uncertainty of the intercept (standard error) as function of the range in $CO_2$ (C) and $\delta^{13}C$-$CO_2$ (D).**



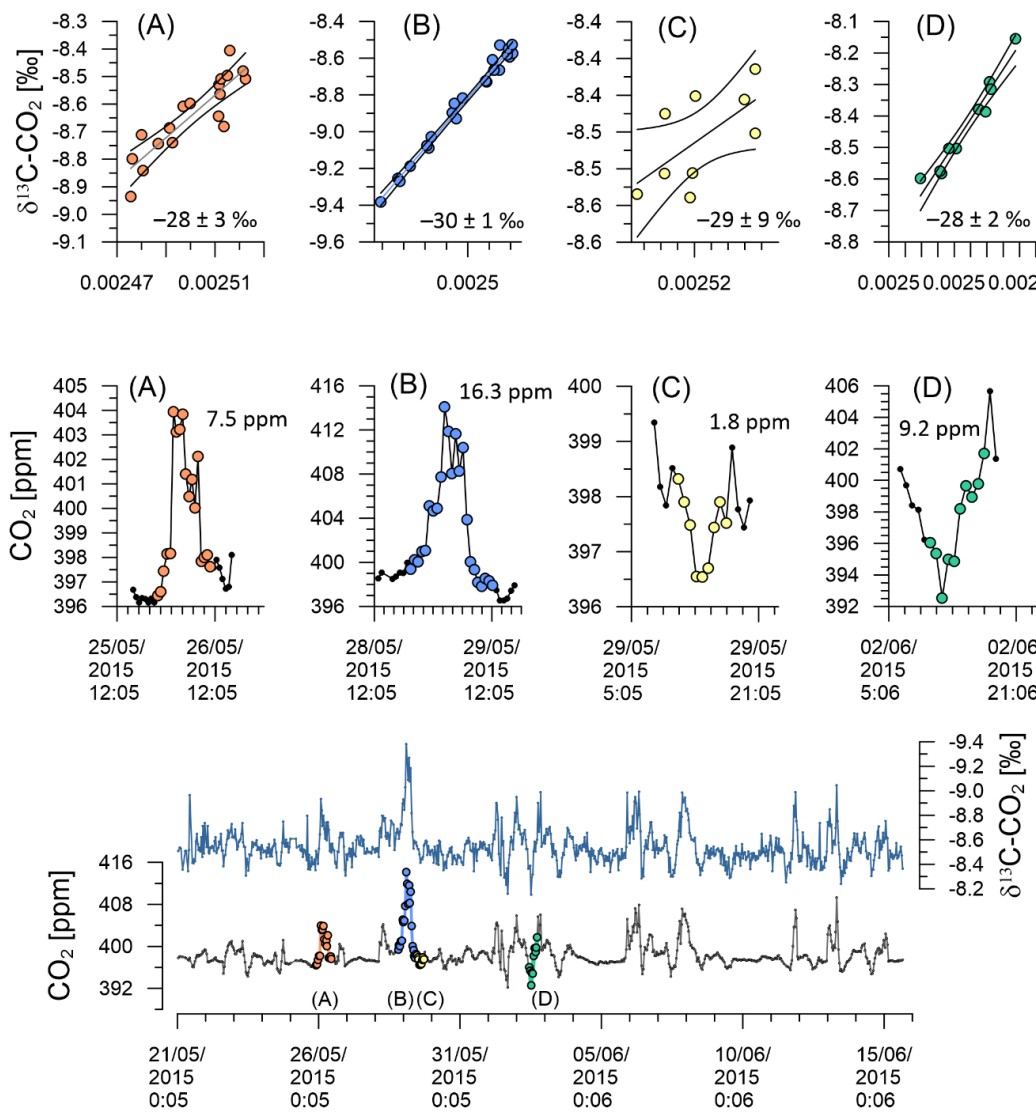

**Figure 16: CO₂ and $\delta^{13}$C-CO₂ time series with selected events (A-D) shown in colours on the CO₂ time series (bottom panel). CO₂ variation of events A-D (middle panel). KPA with 95 % confidence interval and standard error for the intercept of A-D (top panel).**

1275



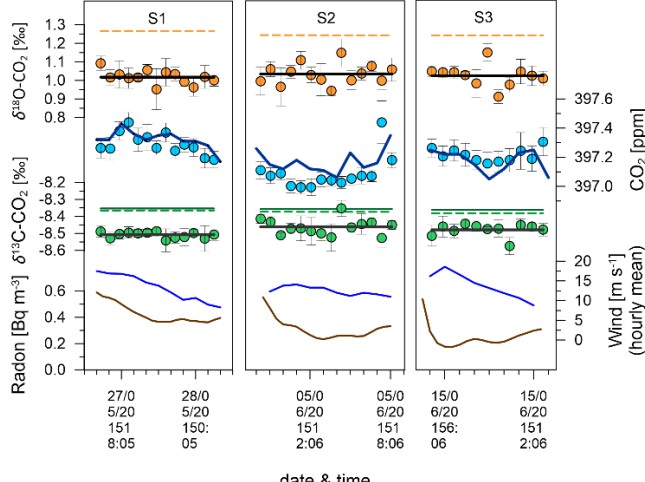

**Figure 17: Data from southerly events S1 (left), S2 (middle) and S3 (right). Radon is shown in brown lines, mean hourly wind speed in blue lines. Green, blue, and orange filled circles show average Delta Ray values for $\delta^{13}C$-$CO_2$, $CO_2$ mole fractions and $\delta^{18}O$-$CO_2$, respectively. Error bars indicating the standard deviation (1 σ). Flat black lines indicate the event averages of the Delta Ray values for $\delta^{13}C$-$CO_2$ and $\delta^{18}O$-$CO_2$. The dark green line shows the interpolated $\delta^{13}C$-$CO_2$ value measured in flask samples before and after the test campaign, while dashed lines represent the interpolated values from $\delta^{13}C$-$CO_2$ and $\delta^{18}O$-$CO_2$ observations at Cape Grim observatory. The dark blue line shows the $CO_2$ mole fraction values as interpolated from 5 min averages of the Siemens analyser at BHD.**