# Peer review of "IRIS analyser assessment reveals sub-hourly variability of isotope ratios in carbon dioxide at Baring Head, New Zealand's atmospheric observatory in the Southern Ocean"

_Atmospheric Measurement Techniques, 2021_

## Referee Comment (RC1)

The authors demonstrated the application of a laser-based instrument to measure the mole fraction and isotopic composition (d13C and d18O) of CO2 in the field. The authors address the calibration protocols and the effect of matrix effect in the air used to dilute pure CO2 in detail. They classify and identify the airmass that reaches their measurement station based on the metrological data (wind direction), radon measurement, and HYSPLIT back tractor. The main focus of the manuscript is on the characterization and calibration of the Delta ray IRS. My main concern as stated in the manuscript, Thermo already ceased manufacturing such an instrument, as a result, it will be important how the calibration and performance evaluation used in this manuscript for Delta ray IRS will benefit other laser-based instruments that measure mole fraction and isotopic composition (d13C and d18O) of CO2.

General comments

1. Arranging the order of figure numbers as they appear in the text will increase the readability of the manuscript
2. In some places there are typos for instance "Line 66: - … deployed the then… "then should deleted

Specific comments

Line 16: - they achieved a precision of 0.07 and 0.06 per mill for d13C and d18O respectively, within the WMO range (again 0.1 per mill). However, online 25 you mentioned a different precision for the WMO network compatibility goal. The difference is not clear, it is also not explained in the main body of the manuscript. This requires attention in the main body of the manuscript since your goal is investigating the capability of Delta ray IRS to achieve the precision recommended by WMO (this is also mentioned in the conclusion section)

Line 39: …. ratio ratios in CO2? delete ratio, maybe even you could rephrase it as… similarly, d18O-CO2 have been used ……
Line 42: - GPP estimates ◊ GPP estimate, and if you include the number 30%, it is necessary why a d18O-CO2 based estimate is higher than the previous GPP values?
Line 65: - the production of the instrument is discontinued by Thermo
Line 66: - … deployed the then… remove then
Line 67: - ….to resolve variation ranges in both…. Should read ….to resolve variations in both…
Line 136: - what is the precision of the mole fraction with the Picarro and with GC since you are measuring with a precious of 0.07 ppm for the CO2 mole fraction using the Delta Ray IRS
- Name and address for Picarro company
- Name and address of the IRMS Company
Table 1:
- why for Kapuni and Marsden you did not use +/- for the uncertainty of delta 13C and d18O values where you give +/- for other gases
- The uncertainty for QC-5 CO2 mole fraction is higher compared to the other standards and it is much higher compared to the precision of the Delta ray IRS
- u_CO2 is better if you define it with parenthesis in the table legend.

Line 170: - How did you come up with a flush time of 150 seconds? Do you already test the memory effect and the optimum value is 150 seconds? It will be good if you add a sentence about how you decided to have a flush time of 150 seconds.

Section 3.3.
- the conversion of transmission to ppm is not clear
Line 70: - This Allan deviation value requires the integration time and the precision of the measurements depends on the integration time the samples are measured?

Section 4.1:
- Allan deviations are dependent for each laser instrument, why the authors assumed the Allan deviation is similar to the previous instrument or is this confirmed by the company
Section 4.2:

- Figure 4:
the figure requires a legend for QC 1 and QC2. For the $CO_2$ mole fraction measurement, for QC with a red marker is more stable, however, we did not see this for the second QC (with a yellow marker).

Line 293: - the second hypothesis, why instability in the referencing step only affects the isotope composition (d13C and d18O) without affecting the $CO_2$ mole fraction?

The paragraph started from Line 330 to 346, for some of the instruments the company name and address are given, however, for some of the instruments it is not provided.
Line 360: Why does the radon measurement contradict the HYSPLIT back trajectory for S1 and S3? It will be good to add a sentence about this difference.

line 380: - Does the instrument measure cell pressure, temp, etc.? I am wondering if a sudden drop in cell pressure might cause such an effect on the isotopic composition?

Line 403: - Why do the cylinders or pressure regulators only affect the mole fraction of $CO_2$?
Line 528: include the error for the average d13C value

Line 530: - To see the negative correlation from figure 11 easily it is better to change the axis label for the d13C from low to high value similar to the $CO_2$ mole fraction (Figure 11 C) and d18O (Figure 11 A)

Figure 14 and in other figures add the error (uncertainty) for the slope and intercept of the linear regression

Line 527: Section 7.3. The message is not clear, the section moves from one reference to another. I recommend reformatting the section: give the summary of the main finding first, then the similarities and differences from other studies.

Line 642: The authors describe an increase in the d18O and d13C of $CO_2$ in nighttime due to respiration and anthropogenic activity. Plant respiration might be a possibility of enrichment when we CO2-H2O exchange with leaf water and soil water assuming the d18O of leaf and soil surface water is enriched compared to the ocean. How combustion would cause an enrichment in d18O of $CO_2$? Paragraph line 637, explains a depletion in the d18O due to

CO2 exchange with depleted water. Is it not in contradiction with the paragraph that started line 642? These arguments need more clarification, maybe using a leaf water record.

---

## Author Comment (AC2)

*General author's comments*

*We would like to genuinely thank the two anonymous referees for their careful review of our manuscript. We absolutely appreciate their time and valuable thoughts in making this a better manuscript! In the sections below, we reply to each of the points made by both referees. We would furthermore like to thank the associate editor Huilin Chen for editorial support.*

*One of the strong concerns of referee #1 was the question if a technical manuscript on an instrument that has been taken off the market is justified. While we totally appreciate this concern, we would like to provide our view at this place in the author's response.*

*Thermo has paused the manufacturing of the Delta Ray. However, already purchased Delta Ray instruments continue to be used by the scientific community and sharing knowledge on how to improve the quality of the measurements remains important. Furthermore, the reason for pausing the manufacturing of the instrument was that the laser manufacturer (not Thermo) stopped making the bespoke lasers. Without the supply of lasers, Thermo was unable to manufacture more instruments. Thermo has looked for alternative suppliers and is potentially starting the manufacturing of the Delta Ray again, if there was sufficient interest by potential users. Therefore, this study provides valuable information for Thermo as well as for current and potential future users alike.*

*Furthermore, we provide a review of published instrument performances of Delta Ray users in the introduction. In our experience, the achievable instrument performance of the Delta Ray is much closer to that of other leading instruments than what is reflected in the literature. We therefore think that this publication is valuable in demonstrating that the capability of this instrument has been underappreciated. Because of this track record, we think it is important to corroborate our finding by providing a greater level of details on laboratory experiments and on observations during instrument deployment than a more typical instrument-test paper might have provided. We strongly believe that this is needed to underpin the suggestion that the performance of the Delta Ray has been underappreciated in the literature and that leaving any of these sections out would not only be a loss for the manuscript, but would bear the risk to significantly weaken our conclusion.*

*We propose that our calibration and QA/QC method provides a more comparable and hence more useful estimation of analytical performance, which further instrument manufacturers as well as users might want to adopt. For example, a common method to determine the instrument performance is using Allan Variance/Deviation plots. While this is very informative on a number of parameters (minimum achievable uncertainty, impact of frequency of calibration on achievable uncertainty, ...), Allan Deviations are specific to the entire analytical system and nature of the sample and are likely to vary with time, laboratory temperature etc. The performance chart method integrates analytical reproducibility over time, including potential states of instrument stability, laboratory temperature cycles, gas changes etc. We trust this is as a more accurate measure of achievable analytical uncertainty and measurement precision of the entire analytical system.*

*For these reasons, we think this publication is justifiable and believe it is in the scope of AMT. However, we do acknowledge that this may be a somewhat uncommon approach for an instrument-test paper and that this adds to a potentially unusual length of the manuscript, as pointed out by referee #2.*

Reviewer #1

The authors demonstrated the application of a laser-based instrument to measure the mole fraction and isotopic composition (d13C and d18O) of CO2 in the field. The authors address the calibration protocols and the effect of matrix effect in the air used to dilute pure CO2 in detail. They classify and identify the airmass that reaches their measurement station based on the metrological data (wind direction), radon measurement, and HYSPLIT back tractor. The main focus of the manuscript is on the characterization and calibration of the Delta ray IRS.

My main concern as stated in the manuscript, Thermo already ceased manufacturing such an instrument, as a result, it will be important how the calibration and performance evaluation used in this manuscript for Delta ray IRS will benefit other laser-based instruments that measure mole fraction and isotopic composition (d13C and d18O) of CO2.

*Please refer to the previous section.*

General comments
1. Arranging the order of figure numbers as they appear in the text will increase the readability of the manuscript

*Agreed. We apologise for the oversight and have enumerated the figures in the sequence of their referral in the text.*

2. In some places there are typos for instance "Line 66: - … deployed the then… "then should deleted

*We apologise for typos we might have overlooked. In this case, however, we purposely wrote about "the then new" instrument, as it was new ~7 years ago, but it is not new anymore.*

Specific comments
Line 16: - they achieved a precision of 0.07 and 0.06 per mill for d13C and d18O respectively, within the WMO range (again 0.1 per mill). However, online 25 you mentioned a different precision for the WMO network compatibility goal. The difference is not clear, it is also not explained in the main body of the manuscript. This requires attention in the main body of the manuscript since your goal is investigating the capability of Delta ray IRS to achieve the precision recommended by WMO (this is also mentioned in the conclusion section)

*Agreed. The description of the two compatibility goal concepts recommended by the WMO is shifted into the main text (introduction) and condensed in the abstract.*

*These compatibility goals were first introduced two decades ago in WMO report 148, following the "CO₂ Experts meeting" in Tokyo. Back then, however, the "network compatibility goal" was referred to as "network precision". The WMO reports are updated every two years, in consent with the experts of the monitoring community. While this concept*

*has been further developed since 2001, the "compatibility goal" concept and values are well-established and often utilised as an agreed data quality objective in publications of the monitoring community. Because it is so well established and well-used, we consider the description and differentiation we provide in the text, along with the reference to the WMO Report 255 as sufficient.*

Line 39: …. ratio ratios in CO2? delete ratio, maybe even you could rephrase it as… similarly, d18O-CO2 have been used ……

*Agreed. Deleted "ratio"*

Line 42: - GPP estimates à GPP estimate, and if you include the number 30%, it is necessary why a d18O-CO2 based estimate is higher than the previous GPP values?

*Agreed. Added "due to shorter cycling time of $CO_2$"*

Line 65: - the production of the instrument is discontinued by Thermo

*Agreed. Changed to "previously manufactured by Thermo"*

Line 66: - … deployed the then… remove then

*See response above to point 2 in "General Comments" section of referee #1*

Line 67: - ….to resolve variation ranges in both…. Should read ….to resolve variations in both…

*Agreed. Changed to "to resolve large variations in both $CO_2$ mole fractions (up to 100 ppm) and isotope ratios (up to 15 ‰)"*

Line 136: - what is the precision of the mole fraction with the Picarro and with GC since you are measuring with a precious of 0.07 ppm for the CO2 mole fraction using the Delta Ray IRS

*Estimates for measurement precisions are provided in Table 1*

- Name and address for Picarro company

*Agreed. Added "(model G2401, Picarro Inc., California, USA)"*

- Name and address of the IRMS Company

*Provided in same paragraph two sentences before*

Table 1:
- why for Kapuni and Marsden you did not use +/- for the uncertainty of delta 13C and d18O values where you give +/- for other gases

*Agreed. Added "±"*

- The uncertainty for QC-5 $CO_2$ mole fraction is higher compared to the other standards and it is much higher compared to the precision of the Delta ray IRS

*True. We stated in line 138 of the initially submitted manuscript, $CO_2$ mole fractions in QC-5 were estimated using the GC-IRMS system, which is not designed for mole fraction measurements. The uncertainty of the $CO_2$ mole fractions in QC-5 is therefore a lot larger than in all other QC gases.*

- u_CO2 is better if you define it with parenthesis in the table legend.

*Agreed. Changed as suggested.*

Line 170: - How did you come up with a flush time of 150 seconds? Do you already test the memory effect and the optimum value is 150 seconds? It will be good if you add a sentence about how you decided to have a flush time of 150 seconds.

*Agreed. The referee #2 has also suggested more clarification would be useful on this matter. Please note that the Delta Ray analyser was provided to us as a loan instrument from Thermo and the time it was available to us was limited. In order to maximise the deployment time at the Baring Head observatory, we did not test the flush time requirements intensively.*

*We suggest changing the text to "We allowed a flush time of 150 s after each gas change, to ensure complete gas replacement in the inlet lines and the optical cell. This was determined by the time it took for stabilisation after switching between two cylinders, plus generous allowance of additional time, with the goal to prevent the need for future adjustments of the measurement sequence. While the flush time could have been optimised further, this was regarded as low priority."*

Section 3.3.
- the conversion of transmission to ppm is not clear

*We appreciate that this is not a straightforward "conversion" from transmission to ppm as the referee suggests. For that reason, we have explicitly stated in section 3.3 that we have no analytical means to make calibrated measurements in the mole fraction range of < 1 ppm. We can only apply an indirect method to estimate $CO_2$ levels. The UHP zero air from BOC*

*has a certified CO₂ mole fraction of < 1 ppm. Assuming the worst case, the UHP zero air from BOC has CO₂ mole fractions = 1 ppm, for which the Delta Ray shows a transmission of 0.7 %. After chemical CO₂ removal, the same UHP zero air from BOC returns a transmission value of 0.2 %. Considering that 1 ppm returns a transmission of 0.7 %, we stated that "**we think that it is likely that**" the resulting transmission of 0.2 % is indicative of a CO₂ blank of < 0.5 ppm, as the transmission is less than half. Figure 2 visualises these results. While this conclusion is clearly not based on calibrated measurements, we believe that the method is sufficiently described and suitable for the purpose of blank assessment. The fact that referee #2 has not commented on this corroborates our position.*

Line 70: - This Allan deviation value requires the integration time and the precision of the measurements depends on the integration time the samples are measured?

*Agreed, Allan Deviation is time dependent. In this introduction, however, our focus is on the achievable minimum value, independent of the time. We have therefore changed the sentence to "Both Van Geldern et al. (2014) and Braden-Behrens et al. (2017) report Allan deviation minima for $\delta^{13}$C-CO₂ of around 0.04 and 0.02 ‰, respectively."*

*We think stating the times of both Allan Deviation minima of both references would defer from the main message and would add length to the manuscript for limited additional value.*

Section 4.1:
- Allan deviations are dependent for each laser instrument, why the authors assumed the Allan deviation is similar to the previous instrument or is this confirmed by the company

*Not sure we fully understand this comment of referee #1. We fully agree that Allan Deviations are instrument specific. We wrote that the Allan deviation minima in the Delta Ray instrument we tested (0.03 ‰ for both isotopes) was comparable to that previously reported for another Delta Ray instrument (Braden-Behrens 2017). We have added "minima" to be more specific. We believe comparing Allan Deviation minima from two instruments is both informative and scientifically sound.*

Section 4.2:
- Figure 4:
the figure requires a legend for QC 1 and QC2.

*Agreed. We have added legends. Thanks to this remark of the referee, we have spotted a mistake in the nomenclature of the gases used for the underlying experiment of Figure 4. The used gases were QC-1 and another test gas, comprising a natural air with slightly higher CO₂ that was not QC-2. Note that the mole fraction and isotope ratios of this gas are not known, but knowing the exact values is not relevant for the conclusion of this test. Because we have not used this gas in any other experiment, we have simply referred to it as "test gas" in the legend.*

For the CO2 mole fraction measurement, for QC with a red marker is more stable, however, we did not see this for the second QC (with a yellow marker).

*We agree there might be a small difference in the amplitudes of the mole fraction measurements in both tanks as observed by referee #1, but this is minute compared to the noise and long-term variability. Moreover, this does not change the conclusion that the mole fraction measurements from both cylinders show a similar pattern and that both do not show a significant shift in mole fractions at the time the isotope measurements in both cylinders shift.*

Line 293: - the second hypothesis, why instability in the referencing step only affects the isotope composition (d13C and d18O) without affecting the CO2 mole fraction?

*This is an interesting question and we have no definite answer. Therefore, we labelled hypothesis ii) as "speculative". It seems, however, plausible for the individual isotope traces to have a small glitch to create a 0.4 ‰ shift on the isotope trace, which is superimposed by the longer-term variability in the total mole fractions, and might not even be significantly.*

The paragraph started from Line 330 to 346, for some of the instruments the company name and address are given, however, for some of the instruments it is not provided.

*Agreed. We have provided details on instrument manufacturers.*

Line 360: Why does the radon measurement contradict the HYSPLIT back trajectory for S1 and S3? It will be good to add a sentence about this difference.

*We do not share the view that the Radon measurements contradict the HYSPLIT back trajectories. Compared to the entire time series, Radon has proven an excellent indicator for potential terrestrial influence within the measured air. This is very consistent with HYSPLIT back trajectories. However, distinguishing Southern Ocean **baseline** air events from more common Sothern Ocean air events is technically challenging.*

*Radon is very low during S1, S2 and S3, suggesting the advected air had very little impact of terrestrial New Zealand during all events. Between the three events, however, Radon is highest during S1, which is in line with the expectations from the HYSPLIT back trajectories. While the Radon levels during S2 and S3 appear similar, with a potentially lower minimum value during some parts of S3, S3 is also marked by higher wind speeds over the entire area, as indicated by the longer length of the back trajectory lines (Figure 13 B and C). The wind speed effect will dilute any potential Radon mixing with the air from the Southern Ocean. Furthermore, Radon is influenced by vertical mixing (not to be confused with the altitude profile of the back trajectory), which the HYSPLIT back trajectories do not indicate.*

*To increase the clarity we made minor modifications and added the following sentence: "Indeed, the Radon levels during S1 are slightly higher than those of S2 and S3, thereby supporting the potential for a small terrestrial component during S1, while the Radon signal during S2 and S3 is indistinguishable."*

line 380: - Does the instrument measure cell pressure, temp, etc.? I am wondering if a sudden drop in cell pressure might cause such an effect on the isotopic composition?

*Agreed. It would be great to have a look into the instrument diagnostics data. Unfortunately, we do not have these data available to us.*

Line 403: - Why do the cylinders or pressure regulators only affect the mole fraction of CO2?

*We never made the general statement that pressure regulators only affect mole fraction measurements. What we did state is that the pattern in the standard deviations of the measurements of isotope ratios on the one hand and mole fractions on the other hand is different. Figure 7 shows that the pattern for mole fractions is clearly specific to each QC gas. We therefore wrote that we think this variability is likely associated with the cylinder or regulator, rather than the Delta Ray itself, because we have no indication for another variable that could cause cylinder-specific noise in the measurements.*

*To clarify this, we changed the sentence to: "The reason for this pattern and the weak performance in $CO_2$ measurements remains unclear, however, we speculate that this is associated with effects in the cylinders or pressure regulators rather than the Delta Ray itself."*

Line 528: include the error for the average d13C value

*Agreed. We changed to " –8.54 ± 0.14 ‰ (average ± 1σ)."*

Line 530: - To see the negative correlation from figure 11 easily it is better to change the axis label for the d13C from low to high value similar to the CO2 mole fraction (Figure 11 C) and d18O (Figure 11 A)

*While we understand the referee's point, we would prefer to keep the figure as is for the following two reasons: 1) not using an inverted scale would require more space between the axes of 11 B and 11 C, which would make the figure a lot larger, 2) features in both records clearly align one way or the other, making this a matter of taste. We prefer the more compact version.*

Figure 14 and in other figures add the error (uncertainty) for the slope and intercept of the linear regression

*We added the uncertainty to all Figures where we thought this is relevant, for example to the Keeling Plots. The current version of Figure 14 clearly delivers the message that the variability of $CO_2$ mole fractions is correlated with variations in $\delta^{13}C$-$CO_2$, while this is not the case for $\delta^{18}O$-$CO_2$. Adding uncertainties to the regression coefficients in the figure will adversely impact on the clarity of the figure, for what we believe has no additional scientific value. We interpret the fact that referee #2 accepted this figure as corroboration for this view.*

Line 527: Section 7.3. The message is not clear, the section moves from one reference to another. I recommend reformatting the section: give the summary of the main finding first, then the similarities and differences from other studies.

*Agreed. This section was lengthy and has more than one objective. We suggest dividing section 7.3 in three parts and included minor modifications to ease the flow between these sections.*

*7.3 $\delta^{13}C$-$CO_2$ observations of the Delta Ray during deployment at BHD*
*7.4 Instrument performance as the limiting factor of the signal size requirement for application of Keeling Plot Analysis*
*7.5 Keeling Plot Analysis using $\delta^{13}C$-$CO_2$ observations from BHD*

Line 642: The authors describe an increase in the d18O and d13C of CO2 in nighttime due to respiration and anthropogenic activity. Plant respiration might be a possibility of enrichment when we CO2-H2O exchange with leaf water and soil water assuming the d18O of leaf and soil surface water is enriched compared to the ocean. How combustion would cause an enrichment in d18O of CO2?

*On this question, we can only refer the referee to the references we provided in this section (Schumacher 2011), which studies the effect of combustion on $\delta^{18}O$-$CO_2$. It would be useful to study combustion products with an in situ isotope analyser within the study region. Unfortunately, this is out of scope/reach, as we do not own such an analyser.*

Paragraph line 637, explains a depletion in the d18O due to
CO2 exchange with depleted water. Is it not in contradiction with the paragraph that started line 642? These arguments need more clarification, maybe using a leaf water record.

*While we agree that leaf water measurements made along the back trajectories at the time the observations were made at Baring Head would be useful, we had no means to generate a leaf water record. Instead, we consulted records of the isotopic composition of precipitation (Baisden 2016).*

*However, we do not see how this is a contradiction. Instead, we interpret this as a proof that the air measured at both times has been influenced by different processes and/or possibly by a different relative contribution of each process during each of the discussed events. These data demonstrate that the Delta Ray system is capable to resolve these as significant differences. Please bear in mind that the observations integrate $CO_2$ processes over many different ecosystems and types of urban sources. In these two events, air masses may have passed over a large range in altitudes (from coast to above tree line), small rural regions where the predominant heat sources are wood burners, to Central Wellington, where natural gas and wood burners are used as main heat source and include further emissions from transport.*

Reviewer #2

The authors tested the performance of an Isotope Ratio Spectrometer to measure CO2, δ13C and δ18O in CO2 in the lab and field. The authors also developed a calibration method for the DR system. This is a carefully done study. The result will be very important to the manufacturer and users.

*We thank referee #2 for this assessment.*

General comments
Although the careful and precise style is important for scientific papers, the thrifty compactness of construction is more necessary, this manuscript is too long.

*Please see our comments on that in the #General author's comments" section.*

The order of the figure numbers should be adjusted as the content.

*Agreed. We apologise for the oversight and have adjusted the order of the enumeration to match the order of appearance.*

At the same time, there are too many figures in the manuscript and some of them are repeated.

*Agreed. We deleted Figure 8 completely, and reduced Figure 16 (now Fig. 15).*

Specific comments
Line149, you mean you applied directly one-point calibration scheme with Q1, and assessed the instrument fluctuations with the results of Q2 and Q3 as target? Did you do some tests with two-point calibration?

*That's exactly what we did. The calibration strategy required by Thermo is based on a two-point calibration, which we found suitable for scale compression control. Unfortunately, it was not possible with the Qtegra version we had to replace the calibration by dilutions of pure $CO_2$ with suitable $CO_2$ in air reference gases.*

Line170-172, Please explain how you decided the flush time and injection time,

*Agreed. Please see response to same comment by referee #1.*

and add the time resolution of the DR

*Agreed. We added: "The time resolution of the instrument is 1 Hz."*

Line306, as the basic introduction of the station, section 5.1 should be brief and general.

*We think it is adequate to introduce the observation systems relevant to this study as well as the site-specific air advection pattern. Because both points are relevant for several sections, we believe this information is best placed in the site description.*

Line347, if possible, section 5.2 can be combined with section 7.2 to help readability.

*Our preference would be to keep 5.2 as a stand-alone section to introduce the basics of wind direction and Radon first, because this is just as relevant for further following sections, not only 7.2.*

Line 392 and Fig. 6, didn't you try to change another test gas? Afterall, Q2 is very important in your scheme with notably different $CO_2$ mole fraction and isotope ratio from Q1 and Q3.

*We fully agree with that assessment and would certainly have done so if we had more time. Unfortunately, the length of time we had the instrument available was very limited, with strict time limit before the instrument got shipped to another institute for testing. The instrument was deployed at Baring Head for 26 days, and it wasn't known at the beginning that QC-2 was unstable. Please also bear in mind that, because this was a borrowed instrument, the data analysis code needed to be written in parallel to deployment, which extended the time until it was clear that QC-2 was underperforming. After finding that QC-2 was unstable, we were unable to prepare another cylinder with natural air and spiked $CO_2$ mole fractions and isotope ratios on time to replace QC-2 by an identical match. This process usually takes a few days for the evacuation of the cylinder and filling with pure $CO_2$ spike, plus time to pump the cylinder and most importantly an additional 2 weeks or more for equilibration. Once the problem with QC-2 was identified, we were lucky to be able to pinch an existing tank from another setup to be used as QC-3. Also, please note that the Baring Head observatory is in a rural location and site visits for maintenance require preparation and travel time.*

Line714, the authors need to provide suitable references or test data for "δ13C-CO2 measurements using air from glass flasks showed that 13C-CO2 was drifting with lowering pressure in the flask".

*We would have liked to be able to do that, however, the instrument failed after we started these tests. We discussed this with Thermo and it is acknowledged that the measurement show a variability that is dependent on the inlet pressure. We have no data to back this up, except a screenshot, which is not suitable for publication. Because this is a significant feature, we would prefer to keep this in the manuscript, even though we have no figure or reference on the magnitude of this effect.*

Fig.4, the authors should provide legends in the figure.

*Agreed. We added legends.*

Fig.10, the middle and bottom panels sent the same information

*The middle and bottom panels in Figure 10 (now Figure 12) also provide additional information that can only be taken from the combination of the panels, i.e. the dependence of statistics on temporal variation of $CO_2$ mole fractions. For example, the standard deviation of the Delta Ray measurements is consistent over the time series, except for times of large mole fraction changes. The latter coincides with an increase of the residuals. We would therefore prefer to keep the panels in this figure. We agree that one of the two $CO_2$ mole fraction time series in the bottom panel could be deleted without significant loss of information, if required.*

Fig.16, the bottom panel of the time series is unnecessary.

*Agreed. We deleted the bottom panel of Figure 16 (now Figure 15).*

---

## Referee Report (RR1)

Line 642: The authors describe an increase in the d18O and d13C of CO2 in nighttime due to respiration and anthropogenic activity. Plant respiration might be a possibility of enrichment when we CO2-H2O exchange with leaf water and soil water assuming the d18O of leaf and soil surface water is enriched compared to the ocean. How combustion would cause an enrichment in d18O of CO2?

There reply:
On this question, we can only refer the referee to the references we provided in this section (Schumacher 2011), which studies the effect of combustion on $\delta$ 18O-CO2. It would be useful to study combustion products with an in situ isotope analyser within the study region. Unfortunately, this is out of scope/reach, as we do not own such an analyser.

Schumacher et al., 2011 reported a combustion experiment for different parts of the plant and also for different plants. However, the maximum d18O value of the combusted CO2 they reported 30 per mill vs VSMOW (look table 3 of Schumacher et al., 2011). How can a 30 per mill will cause an enrichment in the d18O of atmospheric CO2. In their conclusion they mentioned combustion might cause enrichment in the oxygen isotope composition of atmospheric oxygen but not the atmospheric CO2. The enrichment in the d18O of oxygen will arose due to diffusional fractionation during combustion (uncontrolled).

---

## Author Response (AR2)

Dear Editor and referees,

We would like to thank you all for your help to review and improve our manuscript, much appreciated!

As this is the second review and only minor revisions were requested, we will be brief and include our response below.

Line 642: The authors describe an increase in the d18O and d13C of CO2 in nighttime due to respiration and anthropogenic activity. Plant respiration might be a possibility of enrichment when we CO2-H2O exchange with leaf water and soil water assuming the d18O of leaf and soil surface water is enriched compared to the ocean. How combustion would cause an enrichment in d18O of CO2?

There reply:
On this question, we can only refer the referee to the references we provided in this section (Schumacher 2011), which studies the effect of combustion on δ18O-CO2. It would be useful to study combustion products with an in situ isotope analyser within the study region. Unfortunately, this is out of scope/reach, as we do not own such an analyser.

Schumacher et al., 2011 reported a combustion experiment for different parts of the plant and also for different plants. However, the maximum d18O value of the combusted CO2 they reported 30 per mill vs VSMOW (look table 3 of Schumacher et al., 2011). How can a 30 per mill will cause an enrichment in the d18O of atmospheric CO2. In their conclusion they mentioned combustion might cause enrichment in the oxygen isotope composition of atmospheric oxygen but not the atmospheric CO2. The enrichment in the d18O of oxygen will arose due to diffusional fractionation during combustion (uncontrolled).

We gratefully acknowledge that the referee is correct and we have changed the manuscript accordingly to: "It is thus likely that the additional $CO_2$ originated from a combination of anthropogenic sources and ecosystem respiration, and had potentially been subject to exchange of oxygen with a water body that was enriched in $\delta^{18}O$-$H_2O$." We have removed the following sentence, as well as Schumacher et al., 2011 from the reference list.